# De novo-designed transmembrane domains tune engineered receptor functions

**Assaf Elazar[1†], Nicholas J Chandler[2,3†], Ashleigh S Davey[2,3†], Jonathan Y Weinstein[1], Julie V Nguyen[2], Raphael Trenker[2,3], Ryan S Cross[3,4], Misty R Jenkins[3,4,5], Melissa J Call[2,3]\*, Matthew E Call[2,3]\*, Sarel J Fleishman[1]\***

[1]Department of Biomolecular Sciences, Weizmann Institute of Science, Rehovot, Israel; [2]Structural Biology Division, The Walter and Eliza Hall Institute of Medical Research, Parkville, Victoria, Australia; [3]Department of Medical Biology, The University of Melbourne, Parkville, Victoria, Australia; [4]Immunology Division, The Walter and Eliza Hall Institute of Medical Research, Parkville, Victoria, Australia; [5]La Trobe Institute of Molecular Science, La Trobe University, Bundoora, Victoria, Australia

**\*For correspondence:**
mjcall@wehi.edu.au (MJC);
mecall@wehi.edu.au (MEC);
sarel.fleishman@weizmann.ac.
il (SJF)

†These authors contributed
equally to this work

**Competing interest:** See page
24

**Reviewing Editor:** Nir Ben-Tal,
Tel Aviv University, Israel

**Abstract** De novo-designed receptor transmembrane domains (TMDs) present opportunities for precise control of cellular receptor functions. We developed a de novo design strategy for generating programmed membrane proteins (proMPs): single-pass α-helical TMDs that self-assemble through computationally defined and crystallographically validated interfaces. We used these proMPs to program specific oligomeric interactions into a chimeric antigen receptor (CAR) that we expressed in mouse primary T cells and found that both in vitro CAR T cell cytokine release and in vivo antitumor activity scaled linearly with the oligomeric state encoded by the receptor TMD, from monomers up to tetramers. All programmed CARs stimulated substantially lower T cell cytokine release relative to the commonly used CD28 TMD, which we show elevated cytokine release through lateral recruitment of the endogenous T cell costimulatory receptor CD28. Precise design using orthogonal and modular TMDs thus provides a new way to program receptor structure and predictably tune activity for basic or applied synthetic biology.

## Editor's evaluation

This is an interesting article that uses de novo protein design to probe the effects of oligomerization state on the activity of chimeric antigen receptors (CARS). The successful design of transmembrane domains with specific oligomeric states is an impressive result on its own. After experimentally evaluating a couple rounds of designs, the investigators settled on a design protocol that also included screening of the design candidates with docking simulations in alternative oligomerization states to check that the sequences preferred the desired oligomerization state. The designs were experimentally evaluated with gel electrophoresis and X-ray crystallography. In the end, designs that adopted well-defined dimers, trimers, or tetramers were created and carried forward in experiments as CARs.

## Introduction

Interactions among cell-surface receptors play central roles in determining complex structures and controlling signal propagation. In immune receptors (*Berry and Call, 2017*; *Dong et al., 2019*), death receptors (*Fu et al., 2016*; *Pan et al., 2019*), and growth factor receptors (*Arkhipov et al., 2013*;

*Endres et al., 2013*; *Fleishman et al., 2002*), the transmembrane domains (TMDs) govern key interactions involved in assembly, activation, and high-order clustering. Control over the specificity, stability, geometry, and oligomeric state of these interactions is therefore highly desirable both for mechanistic studies of natural receptors and in the engineering of synthetic receptors. Accurate control, however, is difficult to achieve using natural TMDs that have likely been evolutionarily selected for a degree of flexibility in these very attributes (*Matthews et al., 2006*). The importance of high precision in receptor engineering has come into particularly sharp focus with the clinical adoption of cancer immunotherapies using targeted chimeric antigen receptors (CARs) (*Majzner and Mackall, 2019*; *June et al., 2018*; *Salter et al., 2018*; *Eshhar et al., 1993*) to endow T cells with potent antitumor activity. Controlling functional outputs from these engineered single-chain immune receptors poses significant challenges in balancing antitumor CAR T cell activity against toxicities associated with high inflammatory cytokine release, known as cytokine release syndrome (CRS) (*Morgan et al., 2010*; *Gutierrez et al., 2018*; *Morris et al., 2022*; *Maus et al., 2020*).

The modular domain organization of CARs offers a prime example of a synthetic cellular system in which customizable sequences exert control over receptor structure and function. Current-generation CARs comprise an antibody single-chain variable fragment (scFv) domain for tumor antigen binding, a spacer or hinge domain for length and flexibility, a TMD controlling membrane integration and expression levels, and intracellular costimulation and activation domains that provide signals for proliferation, survival, and activation of T cell effector functions. Efforts to imbue CARs with optimal signaling properties have probed all of these domains in one way or another (*Alabanza et al., 2017*; *James, 2018*; *Liu et al., 2015*; *Mata and Gottschalk, 2019*; *Rafiq et al., 2020*; *Balakrishnan et al., 2019*; *Hartl et al., 2020*; *Wu et al., 2020*; *Feucht et al., 2019*; *Majzner et al., 2020*). The TMDs, however, have received little attention in systematic studies of CAR design. For convenience, most incorporate the TMD sequence of the protein from which the adjacent hinge or signaling domains were derived; that is, most commonly from endogenous T cell proteins such as CD4, CD8, CD28, or the T cell receptor (TCR)-associated CD3 ζ chain. At least some of these TMD sequences can engage in molecular interactions that drive self-association and/or assembly with the essential T cell proteins from which they were derived and thereby impact CAR expression and functions in ways that reduce control over signaling outcomes (*Bridgeman et al., 2010*; *Cosson et al., 1991*; *Leddon et al., 2020*; *Call et al., 2006*; *Hennecke and Cosson, 1993*; *Muller et al., 2021*; *Bridgeman et al., 2014*). These may contribute to enhanced function by, for example, driving CAR homodimer formation (*Salzer et al., 2020*; *Bridgeman et al., 2010*; *Fujiwara et al., 2020*), but the involvement of natural immune receptor TMDs in native T cell signaling hampers rational design of CARs with predictable properties.

We set out to define the relationships between TMD structure, CAR oligomeric state, and signaling in CAR T cells by designing completely new TMDs with programmable self-association features and minimal risk of cross-talk with native T cell components. Despite significant recent progress (*Barth and Senes, 2016*; *Korendovych and DeGrado, 2020*), the limitations of membrane protein (MP) atomistic calculations have restricted de novo α-helical MP design studies to highly predictable and rigid coiled-coil motifs (*Joh et al., 2014*; *Lu et al., 2018*) that, while stabilizing them, limited their usefulness as receptor TMDs. By contrast, we recently described an ab initio Rosetta atomistic modeling strategy (*Weinstein et al., 2019*) that uses a new energy function with experimentally determined membrane-solvation terms for each amino acid. This modeling strategy accurately predicts the structure of single-spanning sequences known to self-assemble (*Elazar et al., 2016a*; *Elazar et al., 2016b*). Here, we introduce a new strategy to de novo design programmable membrane proteins (proMPs), resulting in completely new sequences that form TM homo-oligomers of defined geometry and order that can be used to program cell-surface receptor structure. We used these proMPs to generate programmed CAR (proCAR) constructs and found that they endowed T cells with in vivo functional potency that scaled linearly with oligomeric state. proCARs also maintained significantly lower inflammatory cytokine release compared to an otherwise identical CAR containing the natural CD28 TMD, a property that may have safety benefits in clinical applications (*Brudno et al., 2020*; *Ying et al., 2019*; *Morris et al., 2022*; *Rafiq et al., 2020*; *Alabanza et al., 2017*). Our results shed new light on the importance of precision in engineered receptor structure and intermolecular associations for optimal CAR T activity and provide new design tools that may be useful for developing cellular immunotherapies with optimal safety and efficacy profiles.

## Results

### Atomically precise de novo-designed TMDs

In our initial design approach (*Figure 1a*), each design trajectory started from two fully symmetric and extended chains of 24 amino acids encoding either poly-Val or poly-Ala (*Video 1*). In a first, coarse-grained modeling step, backbone torsion angles were sampled from a database comprising three and nine amino acid fragments from α-helical MPs, and the two chains were symmetrically docked against one another with an energy term that disfavored large crossing angles (*Equation 1Weinstein et al., 2019*; *Bowie, 1997*). In a second, all-atom step, we refined the sequence and the structure through iterations of symmetric sequence optimization, backbone minimization, and rigid-body docking using the ref2015_memb atomistic energy function that is dominated by van der Waals packing, hydrogen bonding, and amino acid lipophilicity (*Weinstein et al., 2019*). We noticed that the resulting sequences were overwhelmingly biased towards the large and flexible hydrophobic amino acid Leu (*Figure 1b*), as expected from the dominant role of lipophilicity in the ref2015_memb potential (*Weinstein et al., 2019*). Forward-folding ab initio structure-prediction calculations, however, indicated that the designs were prone to form multiple alternative low-energy dimer structures that were conformationally different from the design conception (*Figure 1—figure supplement 1a*). To mitigate the risk of misfolding due to the high Leu content, we introduced a sequence diversification step comprising 120 iterations of single-point mutation and energy relaxation while biasing the sequence composition to match that of natural TMDs (*Figure 1b*; *Equations 2-3*). The resulting sequences were subjected to ab initio structure prediction calculations (*Das et al., 2009*), and this time, they converged to the design models (*Figure 1—figure supplement 1b*) and exhibited a large energy gap from undesired structures. Previous studies noted that natural TMDs are not optimized for thermodynamic stability (*Faham et al., 2004*). Our design simulations suggest that evolution might have selected sequence compositions to counter TMD misfolding.

Twelve designs were tested in the *Escherichia coli* TOXCAT-β-lactamase (TβL) selection system (*Elazar et al., 2016a*; *Langosch et al., 1996*). In this dual-reporter system, survival on ampicillin and chloramphenicol reports on a design's membrane insertion and self-association propensity, respectively (*Figure 1—figure supplement 2*). Remarkably, most proMPs supported high survival (*Figure 1c*) and two-thirds survived even at the highest chloramphenicol concentration tested, indicating a self-association strength significantly greater than the TMD from the human receptor-tyrosine kinase HER2 (also known as ErbB2), which served as a positive control. Deep mutational scanning of mutant libraries showed that the sensitivity to mutations of most designs was consistent with interfacial versus exposed positions in the design models (*Figure 1d–f*, *Figure 1—figure supplement 3*), suggesting that they indeed assembled through the designed interfaces in the bacterial inner membrane.

Eight proMPs were produced recombinantly as free peptides and all exhibited electrophoretic mobility consistent with SDS- and heat-stable self-association (*Figure 1g*, *Figure 1—figure supplement 3*). The patterns of migration, however, were not uniform. Six proMPs had the apparent molecular weight of a dimer (e.g., proMP 1.5 and 1.6, *Figure 1g*) and exhibited reduced mobility as the peptide concentration was increased, similar to the SDS-stable (*Lemmon et al., 1992*) behavior of the well-studied glycophorin A TMD (*Figure 1—figure supplement 3*). By contrast, the remaining two proMPs exhibited migration patterns that were independent of the sample concentration and had apparent molecular weights more consistent with oligomers larger than the designed dimers (proMP 1.2, *Figure 1g*; proMP 1.3, *Figure 1—figure supplement 3*). To establish the molecular structures of these designs, several were screened for crystallization in monoolein lipid cubic phase, and the structure of proMP 1.2 was determined to 2.55 Å resolution (*Figure 1h*, *Figure 1—figure supplement 4*). While the positions involved in helix packing recapitulated the design model, this proMP indeed formed a trimer instead of the intended dimer, indicating good concordance between mobility in SDS-PAGE and the oligomeric state solved by X-ray crystallography. Ab initio structure prediction calculations in trimeric (C3) symmetry recapitulated the experimentally observed packing interface (RMSD 2.3 Å) (*Figure 1h and i*), demonstrating that it would have been possible to predict this outcome had we considered alternative oligomeric states during design calculations.

Based on this insight, we initiated a third design campaign to produce proMPs in a range of oligomeric states. We incorporated a final step in which ab initio structure prediction calculations (*Weinstein et al., 2019*) were performed in C2, C3, and C4 symmetries for every design. Only those proMPs that were predicted to form the target oligomeric state and none of the alternatives were selected

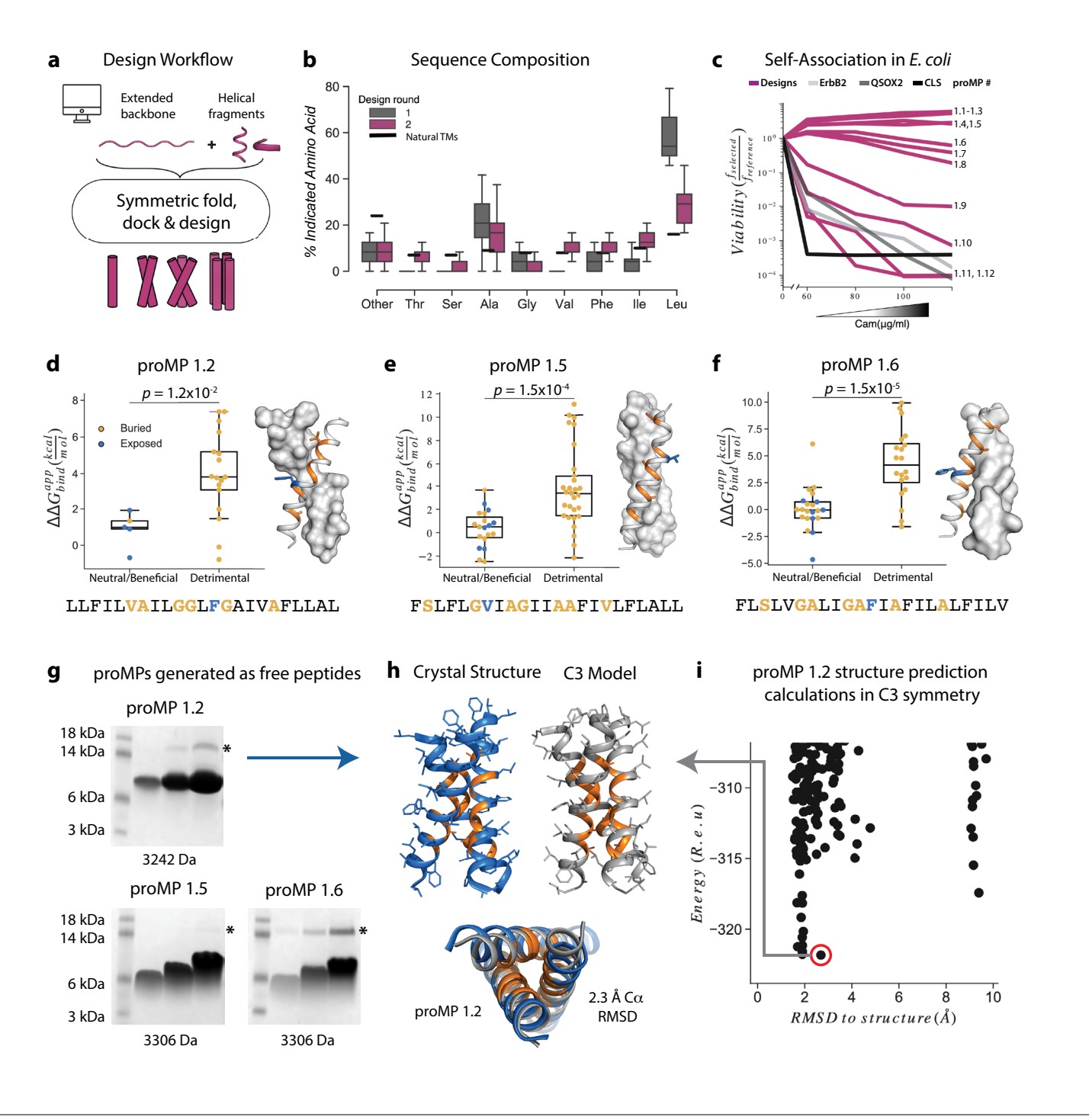

**Figure 1.** Learning the rules for programming self-associating membrane proteins (MPs). (**a**) Rosetta fold, dock, and design uses backbone fragments from natural MPs to construct symmetric, de novo architectures and an MP energy function (*Weinstein et al., 2019*) to optimize the amino acid sequence. (**b**) Round 1 designs were biased towards the hydrophobic amino acid Leu relative to naturally occurring transmembrane domains (TMDs). In round 2, we incorporated a sequence diversification step that conformed the amino acid propensities to those observed in natural TMDs. (**c**) The programmed membrane proteins (proMPs) strongly self-associate in the *E. coli* inner membrane as evidenced by high viability in the deep sequencing TOXCAT-β-lactamase (dsTβL) self-association assay (*Elazar et al., 2016a*). The TMDs from human quiescin sulfhydryl oxidase 2 (QSOX2) and ErbB2 provide positive controls for TMD self-association, whereas the C-terminal portion of human L-selectin (CLS) provides a negative control. (**d–f**) Designed positions that are buried at the interface (orange) are more sensitive to mutation according to dsTβL analysis (*Elazar et al., 2016a*) (*y*-axis) than exposed

*Figure 1 continued on next page*

*Figure 1 continued*

positions (blue). Mutations are predicted to be detrimental or neutral/beneficial using computational mutation scanning of the model structures (Materials and methods). Changes in self-association energies upon mutation are computed according to *Equation 9*. (**g**) proMPs produced as free peptides form SDS-stable homo-oligomers. SDS-PAGE samples containing approximately 15, 45, and 135 µg of peptide were heated to 95℃ for 1 min and run under reducing conditions. * indicates the position of a minor contaminant from the fusion protein used to generate proMP peptides (Materials and methods). Molecular weight below each gel is for a monomer of the corresponding peptide sequence with additional N-terminal EPE and C-terminal RRLC flanking sequences (Materials and methods). See additional examples in *Figure 3*. (**h, i**) The 2.55 Å resolution structure (blue ribbon) determined from crystals grown in monoolein lipid cubic phase (LCP) shows that proMP 1.2, designed to form a dimer, associates to form a trimer in a lipid bilayer environment. (**i**) Forward-folding ab initio prediction of proMP 1.2 in trimeric (C3) symmetry results in a model structure (**h**, gray ribbon) that is very close to the experimentally determined one.

The online version of this article includes the following figure supplement(s) for figure 1:

**Figure supplement 1.** A representative example of programmed membrane protein (proMP) sequence diversification resulting in the sequence for proMP 1.8.

**Figure supplement 2.** The *E. coli* TOXCAT-β-lactamase (TβL) selection system.

**Figure supplement 3.** Additional design round 1 sequences, model structures, gel shift, and deep mutational scanning analysis.

**Figure supplement 4.** Programmed membrane protein (proMP) 1.2 asymmetric unit and structure statistics.

for further analysis (*Figure 2a–d*). This strategy yielded two proMPs for which we obtained crystal structures confirming the target oligomeric state: a dimer with glycine-based packing interface similar to the motif observed in human glycophorin A (proMP C2.1; *Figure 2a–b and e–g*, *Figure 2—figure supplement 1*), and a trimer with an alanine-rich interface that, to the best of our knowledge, is novel in a membrane protein (proMP C3.1; *Figure 2c–d and h–k*, *Figure 2—figure supplement 2*). Interestingly, while two of the helices in the crystal structure of proMP C3.1 aligned well with the design model (*Figure 2i*), the third was in an antiparallel orientation (*Figure 2j*). Despite this arrangement, the six key interface alanine β-methyls were in near-identical positions to their counterparts in the fully parallel model (*Figure 2k*), leading us to suspect that the model is correct but the crystal lattice was enforcing the antiparallel binding mode of the third helix. To probe this possibility, we aligned the parallel model with the asymmetric unit seen in the crystal structure and generated crystallographic symmetry. The resulting model showed clashes for the third helix and indicated that the design model cannot be accommodated in the crystal lattice. The structure thus suggests that this proMP is unintentionally 'reversible' in that one of the helices can form the intended packing mode in either orientation. While this feature is of interest from a design standpoint (*Woodall et al., 2015*), we note that only the fully parallel trimer depicted in the model can form in a biological system where the topology of a single-spanning TMD is constrained by the biosynthetic machinery in a type I orientation.

We conclude that the sequence diversification and the computational selection of the oligomeric state described above provide a practical approach to implement negative design principles that are critical for accurate de novo TMD design (*Fleishman and Baker, 2012*; *Joh et al., 2014*). These new insights will likely also be critical to design de novo hetero-oligomeric TMDs.

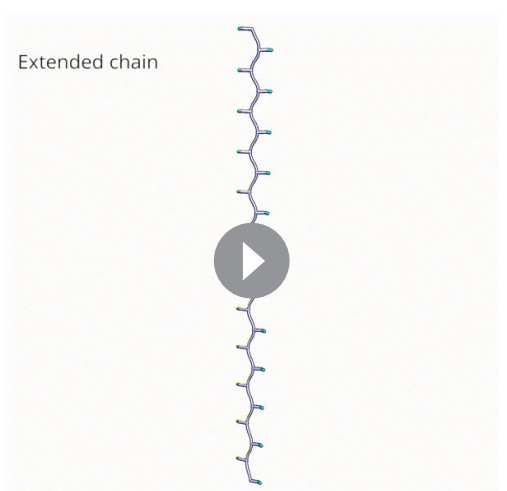

**Video 1.** Key steps in the de novo design of a transmembrane homodimer.
The design process starts from a fully extended chain and uses symmetric fold-and-dock simulations to generate realistic coarse-grained ("centroid" mode) conformations. These are then designed in full atom mode. Finally, a Monte Carlo based sequence diversification step mutates amino acids with a bias towards the sequence propensity of natural amino acids. This diversification step reduces the bias of the full atom design step to Leu amino acids.
https://elifesciences.org/articles/75660/figures#video1

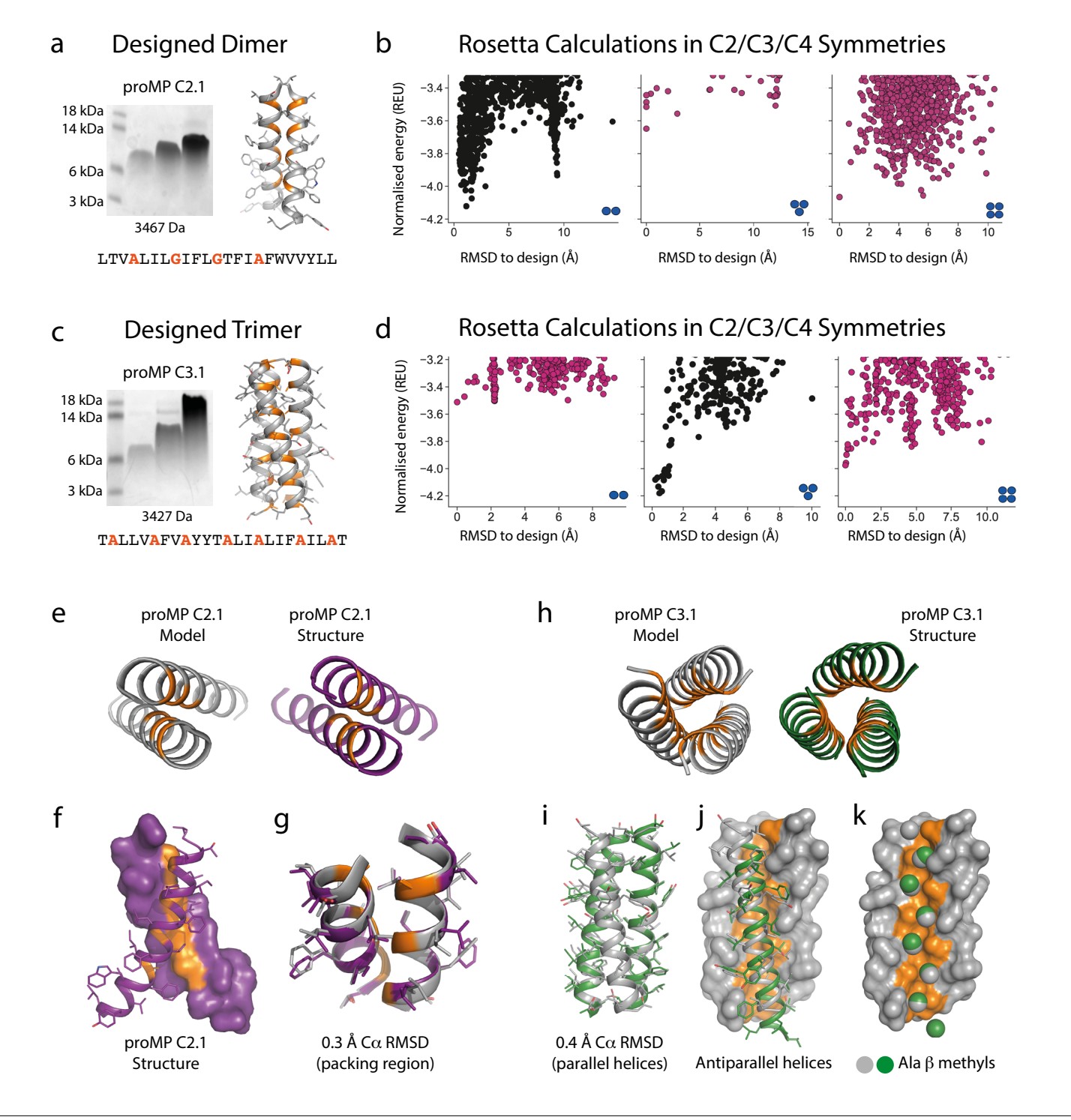

**Figure 2.** Designed membrane proteins (MPs) of defined structure and oligomeric state. (**a**) SDS-PAGE migration of programmed membrane protein (proMP) C2.1 is consistent with a dimer in gel shift assays performed as in *Figure 1*. Design model and peptide sequence shown for reference. (**b**) Rosetta ab initio structure prediction calculations predict that proMP C2.1 preferentially forms a dimer. (**c**) proMP C3.1 exhibits a novel Ala-dominated interface, and its migration pattern at high sample concentration suggests a complex larger than a dimer. Design model and peptide sequence shown for reference. (**d**) Ab initio calculations predict that it primarily forms a trimer. (**e–g**) The proMP C2.1-designed structure is atomically verified by a 2.7 Å crystal structure. Interfacial positions marked in orange. (**h–k**) The crystallographic analysis of proMP C3.1 (3.5 Å resolution) reveals a trimer that is almost identical to the design, although one of the three helices in the trimer is antiparallel relative to the other two in the crystal lattice. Alignment of

*Figure 2 continued on next page*

*Figure 2 continued*

the structure and model (**i**) shows that the antiparallel helix (green) (**j**) positions Ala Cβ methyls that pack into the trimer through the designed interface (gray) (**k**).

The online version of this article includes the following figure supplement(s) for figure 2:

**Figure supplement 1.** Programmed membrane protein (proMP) C2.1 asymmetric unit and structure statistics.

**Figure supplement 2.** Programmed membrane protein (proMP) C3.1 asymmetric unit and structure statistics.

## proCARs with defined oligomeric states

The availability of synthetic TMDs with defined structures provided an opportunity to address two key open questions in receptor engineering: What is the relationship between oligomeric state and functional output? And does the use of natural TMDs impart functional characteristics other than surface localization in the membrane? The hinge-TMD regions in all CARs used in FDA-approved CAR T cell products derive from CD8 or CD28 and drive disulfide-linked receptor homodimer formation (*Fujiwara et al., 2020*). However, the importance of the dimeric state for optimal CAR function is not well understood and alternative oligomeric forms such as trimers or tetramers have not been explored. Furthermore, both CD8 and CD28 TMDs have documented propensities to self-associate (*Hennecke and Cosson, 1993*; *Leddon et al., 2020*). Given the presence of both native receptors in CAR T cells, their use in CARs risks unintended interactions that could affect their expression and/or function. To program CARs that form specific oligomeric states and are insulated from confounding interactions with endogenous signaling proteins, we initially chose the crystallographically confirmed proMPs C2.1 and 1.2 to generate proCARs that form dimers or trimers. These were termed proCAR-2 and proCAR-3, respectively (*Figure 3a*). We also designed a monomeric proMP that exhibited no chloramphenicol survival in deep sequencing TβL (dsTβL) assays (*Figure 3—figure supplement 1*) and used it to produce a monomeric proCAR-1 in order to extend the structure–function study. Our proCAR designs incorporated an anti-HER2 scFv (FRP5; *Wels et al., 1992*) fused to the human CD8α hinge sequence, a proMP-derived TMD, the human CD28 costimulatory sequence, and the human CD3 ζ activating tail. Our reference CAR contained the human CD28 TMD for comparison, approximating a domain configuration that has been extensively studied in vitro and in vivo (*Davenport et al., 2018*; *Davenport et al., 2015*; *Haynes et al., 2002*). In all proCAR constructs, a cysteine residue in the CD8α hinge that mediates disulfide-bonded dimer formation was mutated to alanine (*Figure 3a*) to ensure that the designed TMDs were the primary determinants of oligomeric state.

The HER2 proCARs and reference CD28TM constructs were retrovirally expressed in murine BW5147 thymoma cells. All constructs exhibited similar cell-surface levels (*Figure 3b*), and the reference CD28TM CAR formed disulfide-linked dimers while the cysteine mutant reference (No Cys) and proCARs did not (*Figure 3c*). All CARs were competent to signal when co-cultured with HER2[+] SKBR3 human breast adenocarcinoma cells (*Figure 3d and e*). When expressed in freshly isolated mouse CD8[+] T cells (*Figure 3—figure supplement 2a and b*), all CARs mediated antigen-dependent killing of MC57 mouse fibrosarcoma cells stably expressing HER2 in vitro (*Figure 3f*). Only small differences in killing potency were apparent, with proCAR-1 trending slightly less effective than the reference CARs and proCAR-2 and -3 trending slightly more effective. In vitro cytokine production (IFNγ, IL-2, TNFα, and GM-CSF), on the other hand, was significantly lower in all proCARs, reduced by 2- to 10-fold on average (*Figure 3g*). This effect was not apparent in the CD28TM (No Cys) background and was therefore not due to loss of the CD8α hinge-region disulfide bond.

## The CD28 TMD enhances CAR-mediated cytokine release by associating with endogenous T cell CD28

The striking reduction in cytokine release in all of the proCARs led us to hypothesize that the higher levels of cytokine release in CD28 TMD-containing CARs depend primarily on CD28 sequence features rather than on CAR oligomeric state. The CD28 TMD contains a highly conserved polar YxxxxT motif that is similar to the one that drives CD3 ζ dimerization (*Call et al., 2006*) and is required for optimal dimerization and surface expression of native CD28 (*Leddon et al., 2020*). A recent study showed that the CD28 YxxxxT sequence also causes CARs containing the CD28 TMD to physically associate with the CD28 protein in T cells (*Muller et al., 2021*), but the functional consequences of this association for CAR signaling have not been explored. We modeled this putative CD28TM interface on

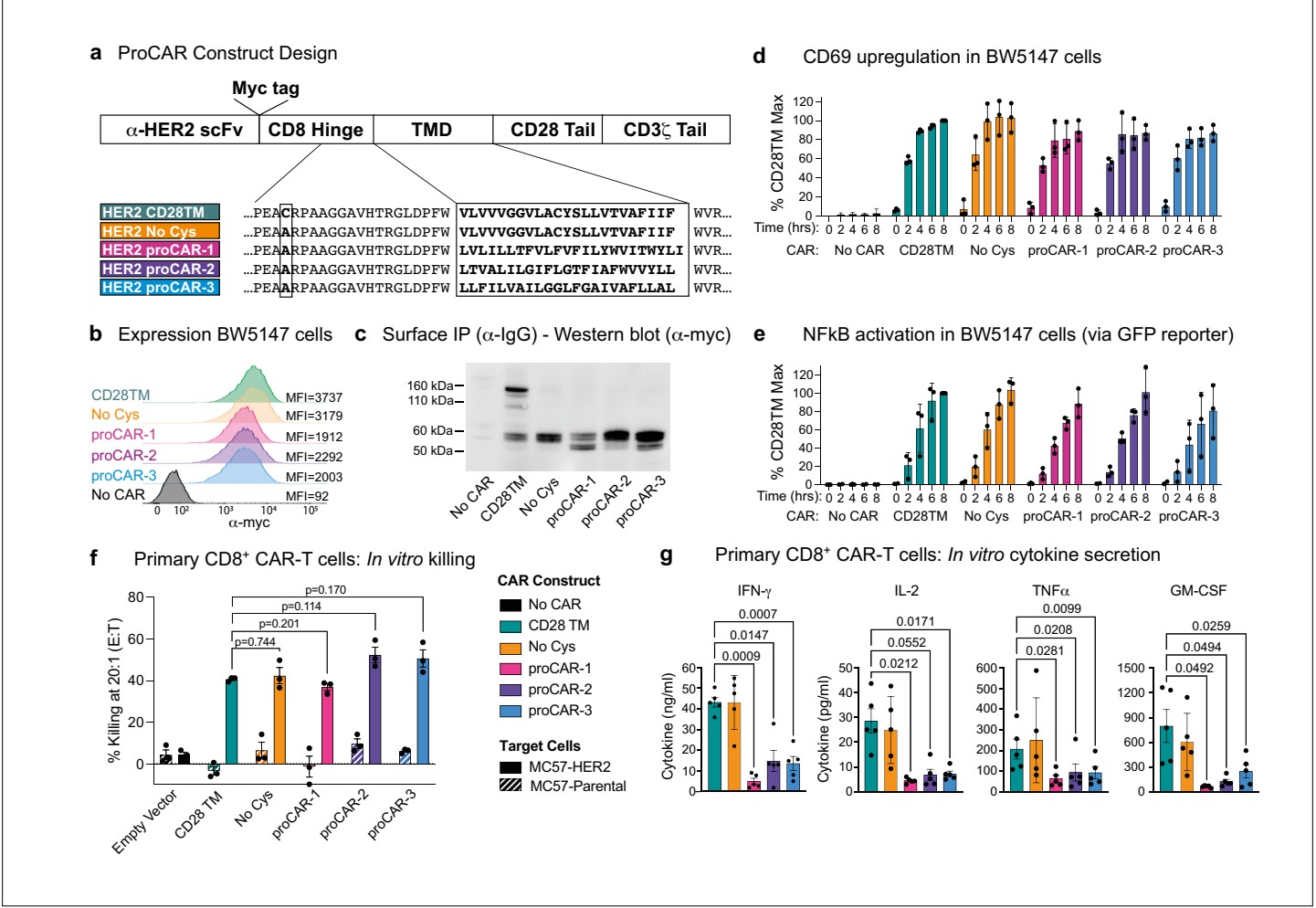

**Figure 3.** Construction and in vitro functional profiling of HER2-specific programmed chimeric antigen receptors (proCARs). (**a**) Schematic showing the domain organization of the reference HER2-specific CAR constructs and modifications made to introduce programmed membrane protein (proMP) transmembrane domains (TMDs). Bold, boxed sequence indicates the human CD28 TMD in the reference CD28TM and no cys CARs and designed proMP sequences in the monomeric (proCAR-1), dimeric (proCAR-2), and trimeric (proCAR-3) receptors. (**b**) BW5147 murine thymoma cells stably expressing proCARs and a destabilized GFP NF-$\kappa$B reporter were surface labeled with anti-Myc antibody and analyzed by flow cytometry to assess surface expression levels. (**c**) Live cells from (**b**) were coated with polyclonal anti-IgG to bind CARs through the scFv domain and immunoprecipitated using protein G beads. Products were separated by nonreducing SDS-PAGE and immunoblotted using anti-Myc antibody to visualize surface-expressed CAR proteins. Molecular weight of the unglycosylated CAR polypeptide is 55 kDa. (**d, e**) Cells from (**b**) were co-cultured with HER2+ SKBR3 human breast adenocarcinoma cells for the indicated times and analyzed by flow cytometry for upregulation of activation marker CD69 (**d**) and GFP expression from the NF-$\kappa$B reporter (**e**). All activation levels are normalized to the 8 hr time point in cells expressing the CD28TM CAR (% CD28TM Max). Bars represent the mean ± SD, and dots show the individual data points for three independent experiments. (**f**) Maximum target killing percentage at 20:1 effector to target ratio from 4 hr $^{51}$Cr release assay. Bars show mean ± SEM with each data point representing an individual experiment (n = 3). p-Values determined from paired t-tests. (**g**) Cytokine production by primary mouse HER2 proCAR T cells following 24 hr co-culture with MC57-HER2 target tumor cells. Bars show mean concentration ± SEM with each data point representing an individual experiment (n = 5). Significance was determined from one-way ANOVA with multiple comparisons. Cytokine production on antigen-negative parental MC57 cells shown separately in *Figure 3—figure supplement 3*.

The online version of this article includes the following figure supplement(s) for figure 3:

**Figure supplement 1.** Design of a highly expressed monomeric programmed membrane protein (proMP).

**Figure supplement 2.** Mouse chimeric antigen receptor (CAR) T cell gating strategy and example programmed chimeric antigen receptor (proCAR) transduction.

**Figure supplement 3.** Raw cytokine secretion against MC57-parental targets for programmed chimeric antigen receptors (proCARs) 1–3.

the $\zeta\zeta$ structure (*Call et al., 2006*) and noted that tyrosine, serine, and threonine in the YSLLVT sequence could all contribute to an interhelical hydrogen-bonding network (*Figure 4a*). We therefore generated a CAR in which this sequence was mutated to **FA**LLV**V**, selectively eliminating the key hydrogen-bonding hydroxyl groups, for comparison to our proCAR and reference constructs. This CAR was well expressed as a disulfide-linked homodimer at the cell surface (*Figure 4b and c*) and generated primary mouse CD8+ CAR T cells whose ability to kill HER2+ target cells in vitro was unimpaired (*Figure 4d*). The CD28TM mutant, however, induced lower levels of cytokine secretion (two- to sixfold lower on average; *Figure 4d*) that were similar to those observed for the proCARs. We therefore concluded that the low-cytokine release seen in the proCAR T cells was likely due to the proCARs being insulated from interaction with endogenous T cell signaling proteins, primarily CD28.

To directly interrogate potential CAR-CD28 associations in primary CD8+ T cells, we examined the four CAR constructs we expect to form dimers in a co-clustering experiment by fluorescence microscopy; these dimers included the CD28TM reference and CD28TM mutant as well as those that ablate the disulfide linkage for comparison (No Cys and proCAR-2). We found that receptors containing the WT CD28 TMD frequently co-clustered endogenous surface CD28, while the CD28TM mutant and proCAR-2 did so significantly less frequently (*Figure 4e and f*). These experiments clearly link the CD28TM interaction motif YSxxxT to high cytokine production in CARs that incorporate this sequence and implicate the recruitment of additional co-stimulatory signaling via endogenous T cell CD28 as the cause. They further substantiate that the de novo-designed TMDs are insulated from these specific interactions.

## In vivo antitumor potency scales with proCAR oligomeric state

Short-term in vitro tumor cell killing assays do not account for variations in proliferation, survival, and cytokine activity that are critical for antitumor activity in a living animal. To evaluate the in vivo antitumor potential of proCAR T cells as a function of receptor oligomeric state, we engrafted NOD-SCID-IL2RG-/- (NSG) mice with the aggressive MC38 mouse colon adenocarcinoma cell line engineered to stably express HER2 and treated them 1 day later with a single intravenous injection of CD8+ CAR T cells (*Figure 5a*). Tumors in mice that received empty vector-transduced T cells grew to ethical endpoint (1000 mm³) within 14 days, while mice that received proCAR-1, -2, and -3T cells slowed tumor growth with potency that increased with oligomeric state (*Figure 5b*, *Figure 5—figure supplement 1*). proCAR-3 provided control that most closely resembled the CD28TM reference CAR T cells. The CD28TM mutant CAR tracked with proCAR-2 and proCAR-3 (*Figure 5c*, *Figure 5—figure supplement 1*), confirming that mutation of the YSxxxT association motif recapitulated the general proCAR functional profile in vivo as well as in vitro. Analysis of mean tumor size at day 14 post-tumor inoculation (the last day all mice were alive) shows a strong inverse correlation with proCAR oligomeric state (*Figure 5d*). These data show for the first time that, all other features being equal, the potency of antitumor CAR T cell activity scales directly with the oligomeric state of the engineered receptor.

## Tetrameric proCAR-4 matches CD28TM CAR tumor control in vivo with substantially lower cytokine release in vitro

This striking correlation between receptor oligomeric state and functional potency prompted us to push the limits further by designing a tetrameric proMP (proMP C4.1), which features extensive alanine-based complementary packing (*Figure 6a*). The free proMP C4.1 peptide migrates on SDS-PAGE predominantly as a single species at a position indicative of a tetramer (*Figure 6b*), consistent with the observation that complementary apolar packing alone can drive stable MP assembly (*Mravic et al., 2019*). HER2 proCAR-4 containing the tetrameric proMP C4.1 TMD sequence was well expressed at the surface of freshly isolated mouse CD8+ T cells (*Figure 6c*, *Figure 6—figure supplement 1*) and supported strong tumor cell killing in vitro (*Figure 6d*). This live-cell imaging assay at low effector:target ratio confirmed that oligomeric proCAR T cells and the T cells expressing the reference CD28TM CAR were all potent killers in vitro, but the monomeric proCAR-1 T cells clearly segregated with weaker killing. Interestingly, the scaled killing we observed in vivo was not evident here, consistent with observations by others that in vitro killing is easily saturated and some differences in CAR activity are only observed at very low target antigen density (*Majzner et al., 2020*). In vivo, proCAR-4 T cells displayed a level of MC38-HER2 tumor control that was indistinguishable from the CD28TM

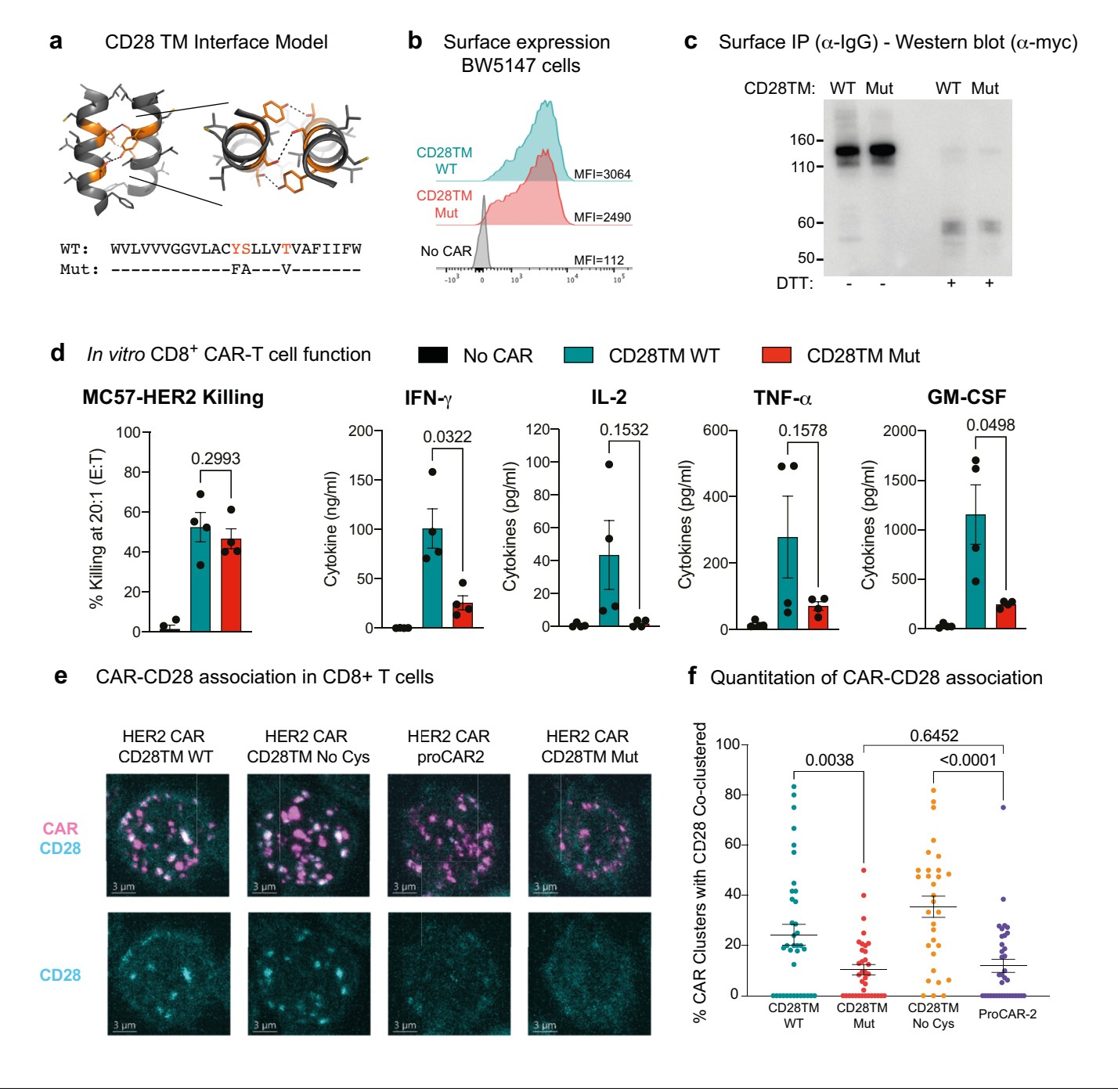

**Figure 4.** Functional consequences of chimeric antigen receptor (CAR)-CD28 association in CAR T cells. (**a**) Model of the CD28TM interface generated by mutagenesis of the CD3ζ TMD (PDB: 2HAC). Polar residues of the CD28 dimerization motif (orange) with predicted hydrogen bonds depicted (dotted lines). (**b**) Surface expression and (**c**) SDS-PAGE and immunoblot analysis of HER2 CARs possessing WT CD28TM or CD28TM mutations depicted in (**a**) expressed in the BW5147 cell line. (**d**) Quantitation of target cell killing measured by chromium release assay and cytokine production by primary mouse CD8[+] CAR T cells in response to the MC57-HER2 target cell line (n = 4). Experiments performed as in *Figure 3*. p-Values determined by paired *t*-tests. (**e**) Representative immunofluorescent confocal images of CAR-CD28 co-clustering in primary mouse CAR T cells. CAR clustering was induced with anti-Myc primary followed by crosslinking with fluorescent secondary antibody (magenta). Cells were then labeled for CD28 (cyan). Images are Z-projections over 12 m, scale bar represents 3 m. (**f**) Quantitation of CAR-CD28 co-clustering, each dot representing the percentage of CAR clusters in one cell that co-localized with a CD28 cluster. Lines show mean CAR-CD28 co-clustering percentage/per cell ± SEM, n ≥ 30 cells. p-Values determined by unpaired *t*-tests.

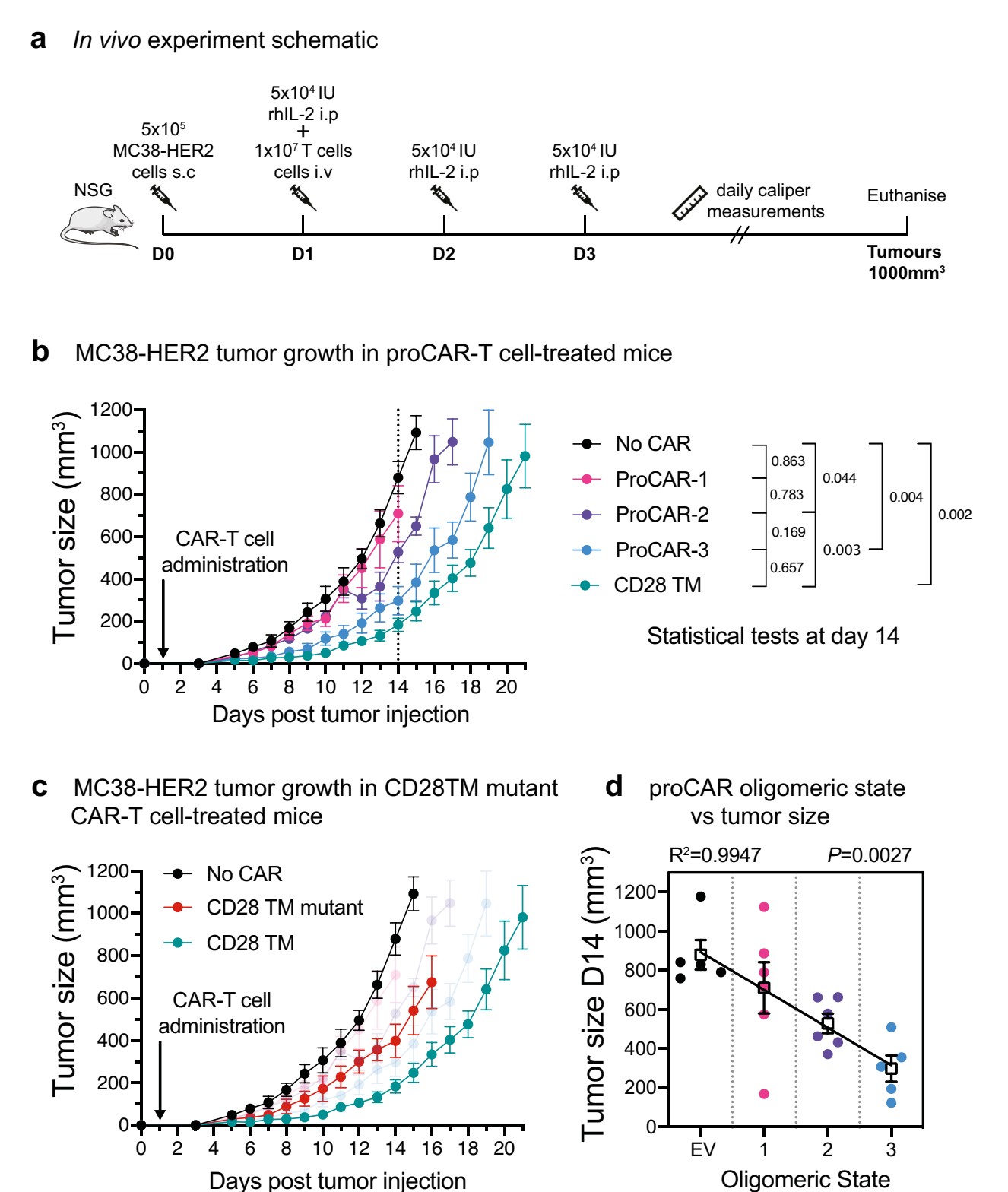

**Figure 5.** In vivo antitumor potency scales directly with programmed chimeric antigen receptor (proCAR) oligomeric state. (**a**) Treatment schedule and experimental setup. NOD-SCID-IL2RG$^{-/-}$ (NSG) mice were injected subcutaneously with MC38-HER2 tumor cells and treated the following day with CD8$^+$ T cells delivered intravenously. Mice were supplemented with daily intraperitoneal injections of recombinant human IL-2 from days 1–3, and tumors measured daily until they reached ethical limits. (**b**) Tumor growth over time for No CAR (empty vector), CD28TM WT, and proCAR T cell groups (n =

*Figure 5 continued on next page*

*Figure 5 continued*

5–6 mice/group). Data points represent mean ± error bars showing SEM. Statistical analysis performed using a two-way ANOVA at day 14. (**c**) Tumor growth over time of the CD28TM mutant group superimposed on (**b**). (**d**) Linear correlation of tumor size on day 14 from (**b**) vs. proCAR oligomeric state, where the '0' point is provided by empty vector (EV)-transduced T cells. Individual data points are colored, mean values in white box and error bars indicate SEM. p-Values indicate the confidence that the slope of the linear regression is nonzero.

The online version of this article includes the following figure supplement(s) for figure 5:

**Figure supplement 1.** Individual tumor growth from in vivo experiment 1.

reference CAR-T cells (*Figure 6e*, *Figure 6—figure supplement 2*), thereby closing the functional gap that was apparent between proCAR-3 and the CD28TM CAR in the previous experiment.

Despite this functional equivalence in vivo, proCAR-4 T cells still released significantly lower levels of cytokines than the CD28TM reference CAR T cells in vitro (*Figure 6—figure supplement 3*). However, the tetrameric design trended towards higher levels of all cytokines than the other proCARs. When we normalized cytokine release to the CD28TM reference across all experiments for all proCAR T cells, the combined data revealed strong linear correlations with receptor oligomeric state for all cytokines tested (*Figure 6f*), reflecting a similar relationship to that identified in the in vivo tumor control data. Taken together, our results reveal that the high cytokine release stimulated by the CD28TM CAR is largely determined by recruiting native CD28 through the TMD. Yet, amongst the proCAR designs that all eliminate this unintended interaction and thereby reduce cytokine release, the relative cytokine levels scale directly with the receptors' oligomeric state. This is consistent with a sensitivity to the number of CAR-encoded CD28 and CD3 tail sequences that can be engaged by a single antigen-binding event. As expected, cytokine production in response to HER2-negative tumor cells was very low in all constructs (*Figure 6g*), showing that pre-assembled higher-order oligomers did not cause spontaneous antigen-independent activation of cytokine production and still required stimulation. These data confirm a robust linear correlation between CAR oligomeric state and CAR T cell functional output, both in vivo and in vitro, that extends at least to the tetrameric state.

## Discussion

This work establishes new de novo TMD design principles that have direct applications in synthetic biology. Starting from a general methodology for the de novo design of membrane-spanning homodimers, we learned that the lowest-energy-designed structures systematically exhibited features that are related to protein misfolding, such as self-assembly through multiple alternative interfaces. Furthermore, biochemical and structural analysis noted a surprising tendency of the designs to self-assemble into higher-order oligomers. To counter these unexpected problems, we developed a new strategy that incorporated negative design principles into an automated design workflow and generated highly expressed and atomically accurate single-span oligomers of defined order. Their formation of SDS-resistant peptide complexes consistent with the target oligomeric state is indicative of high stability. While this analysis does not rule out unintended weak interactions that are disrupted by denaturants, its excellent agreement with design models and experimentally determined structures indicates that the designed interfaces dominate. This paved the way to apply de novo TMD design to the rapidly developing field of engineered receptors, shedding new light on fundamental structure–function relationships in engineered immune receptors.

The outcomes of the proCAR design experiments revealed two specific mechanistic insights into engineered immune receptor function. First, our results highlight how using natural TMDs can confound predictability and control by encoding unexpected functions. In the HER2 CAR used here, CD28 costimulation is explicitly encoded through the CD28 signaling tail incorporated into the CAR protein but is also amplified through a specific sequence signature in the CD28 TMD that recruits endogenous CD28 into activated CAR complexes. A dimeric CAR incorporating the CD28 TMD thus has the potential to recruit two dimers of endogenous CD28, contributing as many as four additional costimulatory sequences to a receptor that carries only two within the CAR sequence itself. This additional costimulatory signaling, which supports both cell division and cytokine production, likely explains a large portion of the enhanced potency and higher toxicity of CD28 TMD-containing CARs compared to those that use TMDs from CD8 or other proteins (*Majzner et al., 2020*; *Fujiwara et al., 2020*; *Brudno et al., 2020*; *Cappell and Kochenderfer, 2021*; *Davey et al., 2020*) and underscores

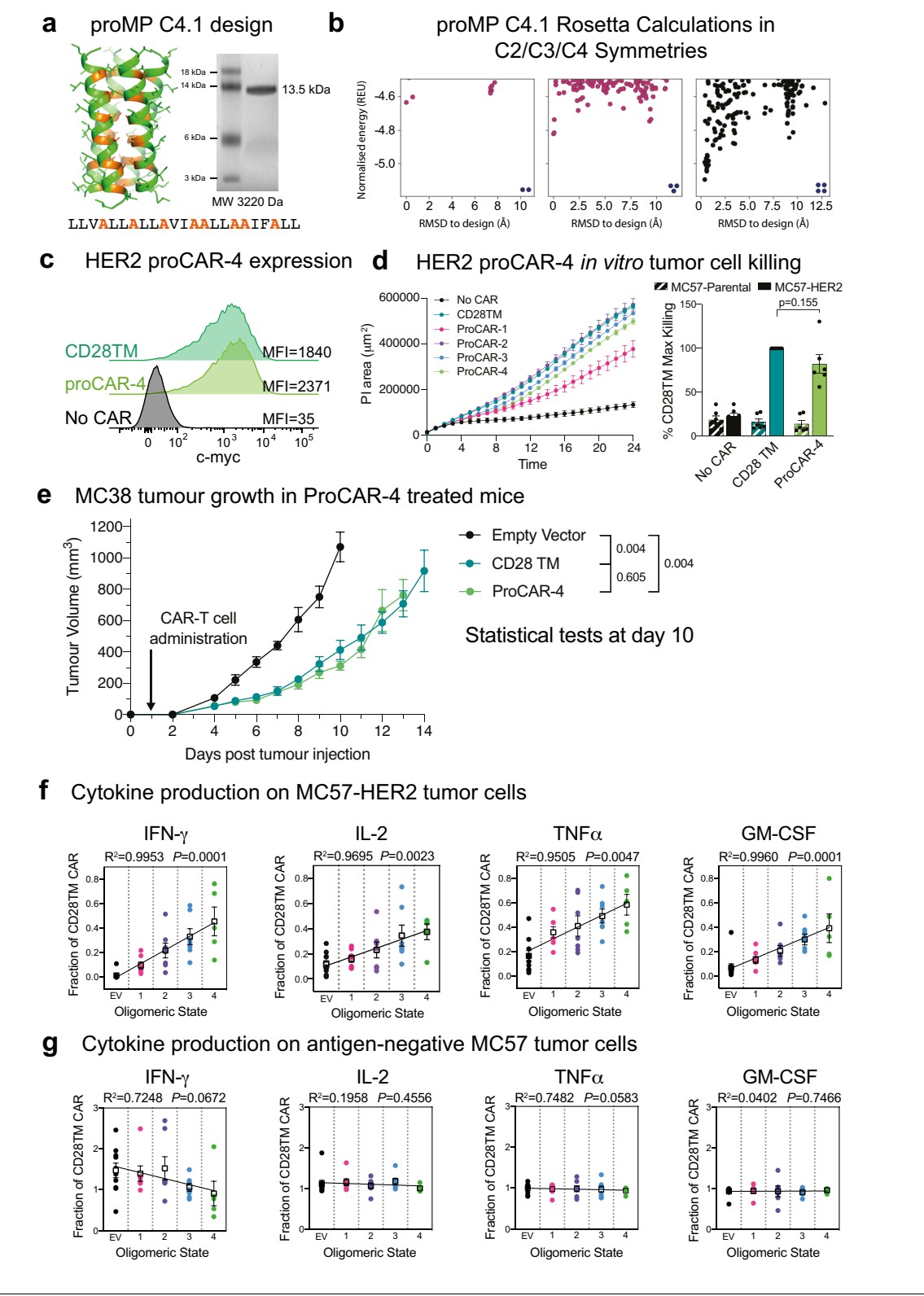

**Figure 6.** In vitro cytokine production scales with programmed chimeric antigen receptor (proCAR) oligomeric state including tetramers. (**a**) SDS-PAGE migration of programmed membrane protein (proMP) C4.1 is consistent with a tetramer. Design model and peptide sequence shown for reference. (**b**) Rosetta ab initio structure prediction calculations predict that proMP C4.1 preferentially forms a tetramer. (**c**) CAR surface expression on primary mouse CD8+ T cells stably expressing CD28TM and proCAR-4 analyzed by c-Myc staining on flow cytometry. HER2 proCAR-4 was designed using the proMP

*Figure 6 continued on next page*

*Figure 6 continued*

C4.1 sequence without the final C-terminal leucine as a transmembrane domain (TMD), inserted as shown in *Figure 3a*. (**d**) IncuCyte killing assay over 24 hr of no CAR, CD28TM, and proCAR1-4 T cells on MC57-HER2 target cells at 1:1 effector to target ratio. Comparison of maximum killing for n = 6 independent experiments shown between CD28TM vs. ProCAR-4. Data points represent individual experiments, with mean ± SEM error bars plotted. (**e**) Tumor growth over time using the same experimental design in *Figure 5a* for No CAR (empty vector), CD28TM WT, and proCAR-4 T cell groups (n = 5–6 mice/group). Data points represent mean ± error bars showing SEM. Statistical analysis performed using a two-way ANOVA at day 10. (**f, g**) Linear correlation of proCAR oligomeric state vs. IFNγ, IL-2, TNFα, and GM-CSF cytokine production (normalized to CD28TM reference) from 24 hr co-culture with (**e**) MC57-HER2 and (**f**) antigen-negative MC57 tumor cells. Individual data points are colored, mean values in white box and error bars indicate SEM.

The online version of this article includes the following figure supplement(s) for figure 6:

**Figure supplement 1.** Example CD28TM vs. proCAR-4 transduction.

**Figure supplement 2.** Individual tumor growth from in vivo experiment 2.

**Figure supplement 3.** Raw cytokine secretion of proCAR-4 vs. CD28TM.

the importance of fully understanding the structure–function relationships in natural TMDs when repurposing them for receptor engineering.

The second major mechanistic insight from this study is that CAR T cell functional potency scales directly with the immune receptor's oligomeric state when all other features are equal. Systematic and robust interrogation of this relationship has never before been possible because type I single-spanning TMDs with well-characterized oligomeric structures are limited (*Trenker et al., 2016*) and functional outcomes that depend strictly on oligomeric state are not easily separated from other features and functions of TMDs (*Bridgeman et al., 2010*; *Bridgeman et al., 2014*; *Wan et al., 2020*). Our de novo-designed proMPs provided a panel of well-characterized, orthogonal TMDs that enabled this finding. The in vitro cytokine production and in vivo tumor control experiments reported here revealed a striking linear correlation between proCAR oligomeric state and the magnitude of T cell responses. Other structural aspects of the receptor complexes, such as the TMD geometry or conformational changes transmitted from the extracellular ligand-binding domains to the intracellular signaling domains, could in principle have roles in receptor activity levels; however, the observed linear relation between oligomeric state and activation suggests that these other aspects play minor roles, if any. Notably, our proMP design workflow can now provide the tools to directly test the role of TMD geometry in CAR signaling by generating different structures of the same oligomeric state for systematic comparisons in future studies.

The ability to broadly attenuate CAR T cell cytokine release while providing a predictable range of functional potencies may have important implications for the development of future cellular immunotherapies. The most effective CAR T cell therapies are accompanied by dangerously high levels of inflammatory cytokine production that cause CRS, which is characterized by fever, hypotension, respiratory distress, and multiorgan failure that can be fatal if not carefully managed (*Morgan et al., 2010*; *Gutierrez et al., 2018*). The current clinical practice is to manage CRS symptoms with cytokine-blocking antibodies and corticosteroids (*Maus et al., 2020*; *Morris et al., 2022*), but approaches to prevent CRS altogether using cytokine gene disruption or modified CAR constructs are areas of active research (*Brudno et al., 2020*; *Ying et al., 2019*; *Sterner et al., 2019*; *Sachdeva et al., 2019*). The lessons we learned from analysis of monomeric, dimeric, and trimeric proCAR designs led to the generation of a tetrameric proCAR with in vivo antitumor activity that precisely matched the potent CD28TM design while still providing a 40–60% reduction in inflammatory cytokine release. Other proCAR designs offer greater reductions in cytokine release but also exhibit concomitant loss of antitumor activity in vivo. The complete proCAR panel thus provides an opportunity to better balance safety and efficacy in CAR T cell therapies by selecting from a spectrum of designs with defined structure–function relationships. Fully assessing their clinical potential will require further testing in mouse models that more closely approximate treatment of established disease in patients, support longer-term persistence studies, and allow direct measurement of cytokine toxicity in vivo.

Importantly, TMD modifications do not directly impact either the antigen-binding or signaling domains. This modularity means that proMP TMDs may be easily implemented on the background of any existing single-chain receptor design and can be combined with other modifications in extracellular or intracellular sequences to expand the combinatorial space available for fine-tuning signaling outputs. This flexibility should facilitate screening for an optimal design for each tumor type and

target antigen. We anticipate that the proMP design methods and sequences will find additional applications for controlling intermolecular cell-surface protein interactions in a variety of synthetic and biological systems.

# Materials and methods

**Key resources table**

| Reagent type (species) or resource | Designation | Source or reference | Identifiers | Additional information |
|---|---|---|---|---|
| Strain, strain background (*Escherichia coli*) | E.cloni 10G | Lucigen | 60107 | High-transformation efficiency electrocompetent cells |
| Cell line (*Mus musculus*) | BW5147 | Kind gift from McClusky lab | CVCL_3896 | |
| Cell line (*Homo sapiens*) | SKBR3 | Kind gift from Jenkins lab | CVCL_0033 | |
| Cell line (*M. musculus*) | MC57 | Kind gift from Jenkins lab | CVCL_4985 | |
| Cell line (*M. musculus*) | MC57-HER2 | Kind gift from Jenkins lab | CVCL_4985 | |
| Cell line (*M. musculus*) | MC38-HER2 | Kind gift from Jenkins lab | CVCL_B288 | |
| Cell line (*H. sapiens*) | HEK293T | Cellbank Australia | Cat# 12022001 | |
| Antibody | Pacific Blue anti-mouse CD69 Armenian hamster IgG monoclonal antibody (clone: H1.2F3) | BioLegend | Cat# 104524 | Flow cytometry (1:200) |
| Antibody | PE anti-mouse CD3ε Armenian hamster IgG monoclonal antibody (clone: 145-2C11) | BioLegend | Cat# 100308 | Flow cytometry (1:800) |
| Antibody | APC-Cyanine7 anti-mouse CD8α rat IgG2a, κ monoclonal antibody (clone: 53-6.7) | BioLegend | Cat# 100714 | Flow cytometry (1:800) |
| Antibody | Alexa Fluor 647 anti-human c-Myc-tag mouse IgG2a monoclonal antibody (clone: 9B11) | Cell Signaling | Cat# 2233S | Flow cytometry (neat) |
| Antibody | APC anti-mouse CD28 Syrian hamster IgG monoclonal antibody (clone: 37.51) | BioLegend | Cat# 102110 | Fluorescence microscopy (1:400) |
| Antibody | Anti-human c-Myc-tag mouse IgG2a monoclonal antibody (clone: 9B11) | Cell Signaling | Cat# 2276 | IP (1:2000) Fluorescence microscopy (1:100) |
| Antibody | HRP anti-mouse IgG polyclonal antibody | Sigma-Aldrich | Cat# A0168 | IP (1:20,000) |
| Antibody | Alexa Fluor 488 goat anti-mouse IgG polyclonal antibody | Abcam | Cat# ab150113 | Fluorescence microscopy (1:200) |
| Antibody | LEGENDplex mouse cytokine panel 2 detection antibodies | BioLegend | Cat# 740149 | (1:4) Reagent used for cytokine detection in **Figures 3, 4 and 6** |
| Antibody | LEGENDplex MU Th panel detection antibodies V02 | BioLegend | Cat# 741045 | (1:4) Reagent used for cytokine detection in **Figures 3, 4 and 6** |
| Antibody | Alexa Fluor647 anti-human CD340 (erbB2/HER2) mouse IgG1, κ antibody (clone: 24D2) | BioLegend | Cat# 324412 | Flow cytometry (1:3200) |
| Recombinant DNA reagent | pMAL-dsTbL (plasmid) | *Elazar et al., 2016a* | Addgene:73805 | TOXCAT β-lactamase assays |
| Recombinant DNA reagent | Designed proCARs | This paper | | See Experimental Methods - DNA sequences of designs |

*Continued on next page*

*Continued*

| Reagent type (species) or resource | Designation | Source or reference | Identifiers | Additional information |
|---|---|---|---|---|
| Sequence-based reagent | Deep sequencing primers | *Elazar et al., 2016a* | | Deep sequencing library preparation and a protocol and analysis as described |
| Peptide, recombinant protein | Retronectin | Takara Bio | Cat# T100B | Final concentration (32 µg/ml) |
| Peptide, recombinant protein | Recombinant mouse IL-2 (ELISA Std.) | BioLegend | Cat# 575409 | |
| Peptide, recombinant protein | Recombinant mouse TNFα (ELISA Std.) | BioLegend | Cat# 575209 | |
| Peptide, recombinant protein | Recombinant mouse IFNγ (ELISA Std.) | BioLegend | Cat# 575309 | |
| Peptide, recombinant protein | Recombinant mouse GM-CSF (ELISA Std.) | BioLegend | Cat# 576309 | |
| Peptide, recombinant protein | Recombinant human IL-2 | PeproTech | Cat# 200-02-1 | Dose ($5 \times 10^4$ IU/injection) Media concentration (100 IU/ml) |
| Commercial assay or kit | Mouse T-activator CD3/CD28 Dynabeads | Gibco | Cat# 11456D | |
| Commercial assay or kit | EasySep mouse CD8a positive kit II | Stem Cell Technologies | Cat# 18953 | |
| Chemical compound, drug | Roswell Park Memorial Institute (RPMI) 1640 Medium +Pen/Strep | Gibco | In-house | |
| Chemical compound, drug | Dulbecco's Modified Eagle Medium (DMEM) | Lonza | Cat# BE12-707F | |
| Chemical compound, drug | Dulbecco's phosphate buffered saline | Gibco | In-house | |
| Chemical compound, drug | Fetal bovine serum | Bovogen Biologicals | Cat# 423101 | Final concentration (10% v/v) |
| Chemical compound, drug | Polybrene | Sigma-Aldrich | TR-1003-G | Final concentration (8 µMg/ml) |
| Chemical compound, drug | Zombie Aqua | BioLegend | Cat# 423101 | Flow cytometry (1:500) |
| Chemical compound, drug | Propidium iodide | Sigma-Aldrich | Cat# P4170 | Final concentration (50 µM) |
| Chemical compound, drug | L-Glutamine | Gibco | Cat# 25030081 | Final concentration (2 mM) |
| Chemical compound, drug | Sodium pyruvate | Gibco | Cat# 11360070 | Final concentration (1 mM) |
| Chemical compound, drug | Non-essential amino acids | Gibco | Cat# 11140050 | Final concentration (1×) |
| Chemical compound, drug | β-Mercaptoethanol | Sigma-Aldrich | Cat# M3148 | Final concentration (50 µM) |
| Chemical compound, drug | Saponin | Sigma-Aldrich | Cat# 47036 | Final concentration (1.2% w/v) |
| Chemical compound, drug | Sodium azide | Sigma-Aldrich | Cat# 71289 | Final concentration (0.1% w/v) |
| Software, algorithm | GraphPad Prism v9.0 | GraphPad Software | | |
| Software, algorithm | FlowJo v10 | FlowJo Software | | |
| Software, algorithm | IncuCyte Analysis Software | IncuCyte Analysis Software | | |
| Other | Rosetta macromolecular modeling suite | Rosetta | Git version: b210d6d5a0c21208f4f874f62b2909f926379c0f | For documentation, see https://www.rosettacommons.org/ |
| Other | $Cr^{51}$ | PerkinElmer | | 100 µCi |
| Other | LEGENDplex mouse IL-2 capture bead A7 | BioLegend | Cat# 740054 | Reagent used for cytokine detection in *Figures 3, 4 and 6* |
| Other | LEGENDplex mouse TNFα capture bead A6 | BioLegend | Cat# 740066 | Reagent used for cytokine detection in *Figures 3, 4 and 6* |
| Other | LEGENDplex IFNγ capture bead A4 | BioLegend | Cat# 740065 | Reagent used for cytokine detection in *Figures 3, 4 and 6* |
| Other | LEGENDplex GM-CSF capture bead B7 | BioLegend | Cat# 740146 | Reagent used for cytokine detection in *Figures 3, 4 and 6* |

## Computational methods

Command lines and RosettaScripts (*Fleishman et al., 2011*) are available in *Supplementary file 1*. Rosetta is available at http://www.rosettacommons.org. We used git version b210d6d5a0c21208f4f8 74f62b2909f926379c0f for all Rosetta calculations.

## Membrane-protein energy function

All atomistic calculations used the Rosetta ref2015_memb energy function *Weinstein et al., 2019*. This energy function is based on the recent Rosetta energy function 2015 (ref2015) energetics, which is dominated by van der Waals packing, electrostatics, hydrogen bonding, and water solvation, with the difference that in ref2015_memb the solvation terms are replaced with splines that recapitulate the amino acid-based lipophilicity contributions observed in the dsTβL insertion profiles (*Elazar et al., 2016a*). The centroid-level energy function was similarly based on ref2015 with amino acid lipophilicity preferences and a biasing potential that disfavors large interhelical crossing angles that are rarely observed in natural TMDs:

$$penalty = 1.51 \times 10^{-4} \times \theta^3 - 8.925 \times 10^{-3} \times \theta^2 + 0.187 \times \theta - 0.532 \tag{1}$$

where θ is the crossing angle between the helix and the membrane normal.

## TMD de novo design

3- and 9-mer backbone fragments were generated for a 24 amino acid poly valine extended chain using the Rosetta fragment picker (*Gront et al., 2011*). The fold and dock protocol was used in all design simulations (*Das et al., 2009*). Briefly, depending on the type of symmetry (C2, C3, or C4), the chains were symmetrically duplicated and each move was applied identically to all chains. Moves included centroid-level fragment insertion and docking, followed by all-atom sequence optimization, and backbone, sidechain, and rigid-body minimization. 50,000 independent trajectories were run and the structure models were filtered using structure and energy-based criteria (the best 1% by system energy, solvent-accessible surface area>700Å; shape complementarity (Sc) > 0.6 [*Lawrence and Colman, 1993*]; ΔΔG$_{binding}$ < –15 R.e.u.; helicality< 0.1R.e.u. [*Weinstein et al., 2019*]). Resulting models were visually inspected and selected for further computational design.

## Sequence diversification

De novo-designed sequences exhibited a high propensity of the amino acid Leu. To reduce this bias, we implemented 120 steps of Monte Carlo simulated annealing sequence design. In each step, a random single-amino acid change was introduced in any position (mutations were restricted to Gly, Ala, Val, Ile, Leu, Met, Phe, Tyr, or Trp). Following relaxation, the mutant was evaluated on three criteria: ΔΔG$_{binding}$, system energy, and the difference between the amino acid propensities in the design versus natural TMDs (*Liu et al., 2002*) using the following equation ($RMSD_{sequence\ comp}$):

$$RMSD_{sequence\ comp} = \sqrt{\frac{\sum_{aa}\left(f\left(aa^{design}\right) - f\left(aa^{natural}\right)\right)^2}{L}} \tag{2}$$

where $f$ is the frequency of a given amino acid, and $L$ is the amino acid sequence length.

The three criteria were then transformed using the 'fuzzy'-logic design sigmoidal function (*Warszawski et al., 2014*):

$$f_x = \frac{1}{1 + e^{(x-o)s}} \tag{3}$$

where $x$ is each of the three criteria, and $o$ and $s$ take the following values: for ΔΔG$_{binding}$ 3 R.e.u. and 1 R.e.u.$^{-1}$, respectively, for system energy 20 R.e.u. and 0.5 R.e.u.$^{-1}$, respectively, and for $RMSD_{sequence\ comp}$ 0.05 and 50, respectively. The $o$ thresholds on binding and system energy were computed relative to the energies of the starting model in each design.

The resulting functions were then integrated into a 'fuzzy'-logic optimization objective function (*Warszawski et al., 2014*):

$$f_{\Delta\Delta Gbinding} \wedge f_{system\ energy} \wedge f_{RMSD_{sequence\ comp}} \tag{4}$$

## Ab initio structure prediction

Designed sequences were subjected to the membrane fold and dock method essentially as described in *Weinstein et al., 2019*. Structure models were filtered using structure and energy-based filters: solvent-accessible surface area > 600 Å; energy < 0; the distance between the TMD ends along the membrane normal, TMsSpanMembrane > 25 Å; fractional agreement between the desired topology for each position (cytosolic, membrane, external) and the designed topology SpanTopologyMatchPos > 0.1.

To evaluate whether the ab initio structure predictions are funneled, we computed the *Z*-score:

$$Z = \frac{E_{lowest} - \bar{E}}{STD(E)} \tag{5}$$

where $E_{lowest}$ is the lowest-energy model with an RMSD of less than 2 Å to the original design model, and *E* represents energies of models with an RMSD > 2 Å and less than 50 R.e.u. from $E_{lowest}$. A cutoff of *Z* > 2.5 was typically used to determine whether an energy landscape was funneled.

## Rosetta mutational-scanning calculations

In order to characterize the effects of mutations on the designs' binding energy, we conducted computational mutation scanning using the FilterScan protocol in RosettaScripts (see XMLs section below). If the difference in total energy for a mutation was >2.5 R.e.u., the mutation was predicted to be detrimental, otherwise it is defined as neutral/beneficial.

## Experimental methods

### TOXCAT β-lactamase assays

DNA encoding the designs and controls were cloned into the pMAL_dsTβL vector (*Elazar et al., 2016a*) (available at AddGene #73805) using *XhoI* and *SpeI* restriction sites and selected by growth on spectinomycin and ampicillin in standard concentrations. For positive controls, the natural ErbB2 and QSOXS2 TM domains were chosen (representing strong and weak homo-oligomers, respectively; *Schanzenbach et al., 2017*). The monomeric C-terminal portion of human L-selectin (CLS) (*Srinivasan et al., 2011*; *Elazar et al., 2016a*) was chosen as a negative control. Resulting plasmids were transformed into *E. coli* cloni cells (Lucigen), plated on agar plates containing 50 μl/ml spectinomycin followed by overnight growth in a 37°C at 200 rpm. Cultures were then inoculated into fresh LB + 50 μl/ml spectinomycin medium to $OD_{600}$ 1 and then plated on Petri dishes containing 50 μl/ml spectinomycin, 100 μl/ml ampicillin, or 100 μl/ml ampicillin with a range of different chloramphenicol concentrations. For single-clone growth assays, 2 μl of cultures at OD 0.1 were diluted and plated on square Petri dishes containing different chloramphenicol concentrations (extended data in *Figure 1—figure supplement 2*).

### Deep sequencing analysis

A library encoding all of the designed sequences, controls, and single-point mutations in defined positions (using NYS codons to encode hydrophobic and small, mildly polar amino acids) was transformed and grown in large 12cm Petri dishes on different chloramphenicol concentrations (0, 60, 80, 100, and 120μl/ml for data in *Figure 1C* and extended *Figure 3—figure supplement 1* and 0, 21, 27, 34, 42, 52, 66, 82, 102, 128, 160, and 200 μl/ml for data in *Figure 3—figure supplement 1C*) overnight. Chloramphenicol concentration of 60 μl/ml was selected for the analysis for *Figure 1D-F* and *Figure 1—figure supplement 3*. Bacteria were harvested and subjected to deep sequencing library preparation and a protocol and analysis as described in *Elazar et al., 2016a*.

### Deriving changes in free energy of self-association from the deep mutational scanning data

From the deep sequencing analysis, we compute the propensity *p* of each mutant *j* at position *i* relative to the wild type as described in *Elazar et al., 2016a*:

$$p^{i,j} = \frac{count^{i,j}}{count^{wt}} \tag{6}$$

where *count* is the number of reads for each variant, adding a pseudo-count of 1 if no reads were detected for the wild type. We then obtain selection coefficients *s* by comparing the selected and reference populations:

$$s^{i,j} = \frac{p_{selected}^{i,j}}{p_{ref}^{i,j}} \tag{7}$$

where the selected population is selected on ampicillin + chloramphenicol plates (selection for insertion and self-association, respectively) and the reference population is selected only on ampicillin plates (insertion only). At each position *i*, the selection coefficients are transformed to changes in free energy of self-association from the wildtype identity *wt* to the single-point mutation *j* through the Gibbs free-energy equation:

$$\Delta\Delta G_{i,wt \to j}^{measured} = -RTln\left(\frac{s^{i,j}}{s^{i,wt}}\right) \tag{8}$$

where *R* is the gas constant, *T* is the absolute temperature (310 K), and *ln* is the natural logarithm.

In the TOXCAT-β-lactamase construct, bacterial viability on chloramphenicol depends on the activity of the ToxR chloramphenicol acetyltransferase moiety, which in turn depends on oligomer concentrations (*Langosch et al., 1996*; *Russ and Engelman, 1999*; *Elazar et al., 2016a*). Oligomer concentrations depend on both membrane insertion and self-association energy (*Duong et al., 2007*; *Elazar et al., 2016a*). Therefore, the energy computed in *Equation 8* comprises contributions from both membrane insertion (doubled in the case of homodimers) and self-association energy. Thus, to extract the self-association energies for each point mutation, the apparent free energy of self-association subtracts the apparent contribution from insertion:

$$\Delta\Delta G_{i,wt \to j}^{app,assoc.} = \Delta\Delta G_{i,wt \to j}^{measured} - 2\Delta\Delta G_{i,wt \to j}^{app,ins.} \tag{9}$$

where the apparent change in free energy of insertion is computed according to the per amino acid, membrane depth-dependent insertion energies derived from the dsTβL assay in *Elazar et al., 2016a*.

## DNA sequences of designs

>proMP 1.1
CCTTTATCTTTCCTCTTAGGGATACTAGCTGCGCTGGTGGGGTTCATCATTGGCTTTTTAGCGG
CCTTGATT
>proMP 1.2 (trimer; used in proCAR-3)
CCTTTGTTATTTATTCTCGTCGCAATACTTGGAGGCTTATTTGGGGCGATTGTTGCATTCCTTT
TGGCGTTA
>roMP 1.3
CCGATCCTGTTCGCAATACTGGCGGCTTTCATCGGGGGCATTTATAGCTGCCCTGTTCGTG
CTAGTATTGGCA
>proMP 1.4
CCCTTTGGAGCTTTACTAGCAATCATAGCATTCGTCGTAGGAATGTTATTCTCAGCATTCGTTT
TACTCATC
>proMP 1.5
CCCTTTAGCTTGTTTTTGGGCGTTATAGCCGGCATTATTGCTGCATTCATCGTTTTATTCCTGG
CATTACTA
>proMP 1.6
CCTTTTTTATCGCTTGTTGGTGCGCTAATCGGGGCTTTCATAGCATTTATCTTGGCTTTGTTCA
TTTTGGTT
>proMP 1.7
CCGATTCTGATCACTTTGGCAATGCTTACGGGAGCAGTGATTGGGGCGATCTCGTCTTTT
CTCCTAGTGTAT
>roMP 1.8
CCAGCCTTTTATATTATATTGGCAATTCTCACCTCGTTCATAGCCTATTTGGTGGGTCTACTCG
TGTCTTTT
>proMP 1.9

CCTATTTACGTTATACTAGCCATCTTGGCCGCGGTATTCACTTGGTTCATAGTCCTTATAACTA
GCCTGAGT
>roMP 1.10
CCTACGGTTACGAGTGCGATTCTTGGCGTGTCATTCGGTACCTTTATTAGCCTCGTAGCT
CTGTGGCTTGCA
>proMP 1.11
CCAGTGATTGCAATCTTAACTTTTATAGTCCTCACTGCGATTTCGGGAGCGCTGCTCGCT
GTTTGGTTCTCC
>roMP 1.12
CCCATCGTCTTGCTCCTCAGTCTACTCGCCAGTGTATTTGGGGCGTTCATCACATTTATT
TGGGCTTACTTG
>proMP C1 (monomer; used in proCAR-1)
CTGGTGCTGATTCTGCTGACCTTTGTGCTGTTTGTGTTTATTCTGTATTGGGTGATTACCTGGT
ATCTGATT
>proMP C2.1 (dimer; used in proCAR-2)
CCGCTGACCGTGGCGCTGATTCTGGGCATCTTCCTGGGCACCTTTATTGCGTTTTGGGTG
GTGTATCTGCTG
>proMP C3.1
ACCGCGCTGCTGGTGGCGTTTGTGGCGTATTATACCGCGCTGATTGCGCTGATTTTTGCG
ATTCTGGCGACC
>proMP C4.1 (tetramer; used in proCAR-4)
CCCCTTTTAGTCGCCTTATTGGCGCTGCTTGCTGTAATCGCCGCATTATTAGCAGCTATCTTTG
CATTGCTG
>CLS
CCGCTGTTCATCCCGGTTGCAGTTATGGTTACCGCTTTTAGTGGATTGGGGTTTATCATC
TGGCTGGCTAC
>ErbB2
TCTATCATCTCTGCGGTGGTTGGCATTCTGCTGGTCGTGGTCTTGGGCGTGGTCTTTGGC
ATCCTGAT
>QSOX2
AGCCTATGCGTTGTTTTATACGTGGCATCTAGTTTATTTATGGTCATGTACTTCTTC

## proMP peptide production

Peptides were produced recombinantly as 9His-trpLE fusion proteins in *E. coli* following a previously published protocol (*Sharma et al., 2013*). To aid purification, analysis, and crystallization, all designed sequences were modified to include Glu-Pro-Glu at the amino terminus and Arg-Arg-Leu-Cys at the carboxy terminus based on the favorable properties of the glycophorin A TMD fragment whose structure has been previously determined by X-ray crystallography (*Trenker et al., 2015*). Dissolved fusion protein from inclusion bodies was purified on nickel affinity resin, cyanogen bromide digested, and reverse-phase HPLC purified following the published procedure (*Sharma et al., 2013*) with the following modifications: the C-terminal Cys sulfhydryl group was protected using 10 mM S-methyl methanethiosulfonate (MMTS, Sigma-Aldrich) during lysis and inclusion body solubilization and peptides were at no time disulfide linked. HPLC-purified peptides were stored as lyophilized products at room temperature (RT) until needed.

## SDS-PAGE analysis

Samples were prepared by drying indicated amounts of each purified peptide taken from dried and weighed product redissolved in 1,1,1,1,1,1-hexafluoroisopropanol (HFIP, Merck). Samples were lyophilized, redissolved in 25 µl 1× NuPAGE LDS sample buffer (Thermo Fisher Scientific), and heated for 1 min at 95°C. Cooled samples were separated on 12% NuPAGE Bis-Tris gels (Thermo Fisher Scientific) at 200 V for 40 min and visualized by staining with Coomassie Blue R-250 (Bio-Rad).

## Crystallization screening and structure determination

### proMP crystallization in LCP

For reconstitution into LCP, lyophilized peptide was weighed and co-dissolved with appropriate amounts of monoolein (Nu-Chek Prep) in HFIP. Solvent was removed under streaming nitrogen, followed by lyophilization overnight. Peptide-monoolein mix was heated (52°C) until liquid and mixed

3:2 with 10 mM Tris pH 8.0 for LCP formation using coupled 100 µl gastight Hamilton syringes (Formulatrix) at RT. For screening, LCP mixture was dispensed in 100 nl drops onto 96-well glass plates (Molecular Dimensions) with 1000 µl of precipitant solution using a Mosquito LCP robot (TTP Labtech) at RT. Plates were sealed and kept at 20°C in a Rock Imager 1000 (Formulatrix) for incubation and monitoring of crystal formation.

### proMP crystalliszation in detergent

For reconstitution with detergent, lyophilized peptide was weighed and dissolved in 30 mM detergent ($C_8E_4$; Anatrace, $C_8E_5$; Anatrace) in HFIP. Solvent was removed under streaming nitrogen followed by lyophilization overnight. Peptide-detergent mix was reconstituted in 10 mM Tris pH 8.0. For screening, peptide-detergent mixture was dispensed in 150 nl drops onto SD-2 plates (IDEX Corp) with 150 nl of precipitant solution using a Crystal Phoenix robot (Art Robins Instruments) at RT. Droplets were equilibrated against 50 nl of crystallant in the reservoir. Plates were sealed and kept at 20°C in a Rock Imager 1000 (Formulatrix) for incubation and monitoring of crystal formation.

### Data collection and structure determination

Data were collected on the MX2 beamline of the Australian Synchrotron at a wavelength of 0.9537 Å and a temperature of 100 K. Data were indexed and scaled using XDS (*Kabsch, 2010*) and Aimless (*Winn et al., 2011*). Structure factor amplitudes were obtained using cTruncate (*Davenport et al., 2015*). 6W9Y was solved with Phaser (*McCoy et al., 2007*) by molecular replacement using the GpA monomer helix as a search model (PDB code 5EH6 *Trenker et al., 2015*). 6W9Z was solved with Phaser by molecular replacement using 5EH6 mutated to the proMP C2.1 sequence in *Coot* (*Emsley et al., 2010*). 6WA0 was solved with Phaser by molecular replacement using the designed trimer as a search model. This resulted in a model that contained good density for two chains, with the final chain of the trimer considerably worse. The third chain was removed and a second molecular replacement job was performed with the first two chains fixed in place and a single helix from the model trimer used as a search model. This resulted in placement of the third helix in an antiparallel direction with respect to the other two chains, and this was judged as correct based on comparison of overall Rfree of each model, average B factors of each chain, and visual inspection of the electron density in *Coot* (*Emsley et al., 2010*). Iterative rounds of refinement and model building were performed in PHENIX (*Liebschner et al., 2019*) and *Coot* (*Dong et al., 2019*).

### proCAR construct preparation

The HER2-specific CAR used was based on a previously described construct (*Haynes et al., 2002*). Restriction digest sites were removed and human sequences were Gibson cloned together and inserted into EcoRI/XhoI digested pMSCV-IRES-mCherry-II vector (NEB Gibson Assembly Master Mix, Cat# E2611L). The CAR construct contains the FRP5 anti-HER2 scFv, Myc tag, human CD8α stalk, human CD28 TM/ tail, and human CD3 ζ tail sequences. PCR primers were used to generate a cysteine to alanine mutation in the CD8α stalk region to prevent covalent dimerization. Overlapping PCR was used to generate CARs with altered TM domains on the background of the cysteine-mutated CD8α stalk. These constructs were inserted into the pMSCV-IRES-mCherry-II vector via EcoRI/XhoI restriction sites.

### Animals

All mice were of an inbred C57B/6J or NOD.Cg-Prkdc$^{scid}$IL2rg$^{tmWjl}$/SzJ (NSG) genetic background. All animal experiments were approved and performed in accordance with the regulatory standards of the Walter and Eliza Hall Institute Animal Ethics Committee (approval: WEHI-2019.020).

### Mouse CD8$^+$ T cell isolation and culture

Single-cell suspensions of peripheral lymph nodes from 6- to 8-week-old C57B/6 mice were prepared by mechanically dissociating through a 70 µm cell strainer (BD Biosciences) into cold phosphate-buffered saline (PBS). CD8$^+$ T cells were subsequently selected using the EasySep mouse CD8a positive Kit II (Stem Cell Technologies) according to the manufacturer's instructions. Purity was confirmed as >95% using LSR II Fortessa (BD Bioscience), FACSymphony (BD Biosciences), or Aurora (Cytek). CD8$^+$ T cells were subsequently activated by incubating overnight with Mouse T-Activator CD3/CD28

Dynabeads (Gibco) at a bead-to-cell ratio of 1:1 in mouse T cell medium (mTCM) consisting of Roswell Park Memorial Institute (RPMI) 1640 Medium (Gibco) supplemented with fetal bovine serum (10%; Bovogen Biologicals), L-glutamine (2 mM; Gibco), sodium pyruvate (1 mM; Gibco), nonessential amino acids (1×; Sigma-Aldrich), β-mercaptoethanol (50 μM; Sigma-Aldrich), and recombinant human IL-2 (100 IU/ml; PeproTech). Following removal of magnetic beads, T cells were maintained at $1 \times 10^6$ cell/ml in mTCM.

## Cell lines

293T, MC57, MC57-HER2, SKBR3, and MC38-HER2 cell lines were cultured in Dulbecco's Modified Eagle Medium (DMEM) supplemented with fetal bovine serum (10%; Bovogen Biologicals) and L-glutamine (2 mM; Gibco), incubated at 37°C, 10% $CO_2$. BW5147 cells were cultured in RPMI supplemented with fetal bovine serum (10%; Bovogen Biologicals) and L-glutamine (2 mM; Gibco), incubated at 37°C, 5% $CO_2$. Cell line identity was not independently verified by genetic testing. HER2 expression on tumor target and control cell lines was confirmed via anti-HER2 surface staining (BioLegend, Cat# 324412) and flow cytometry. All cell lines were regularly confirmed mycoplasma negative using the Stratagene Mycosensor PCR Assay Kit (Agilent, Cat# 302108).

## BW5147 and primary mouse CAR-T cell generation

Retrovirus for all T cells was produced using calcium phosphate transfection of HEK293T cells. BW5147 cells expressing a destabilized-GFP NFkB reporter element were mixed 1:1 with filtered viral supernatant at a final density of $2.5 \times 10^5$ cells/ml. Polybrene transfection reagent (Merck) was added to a final concentration of 8 μg/ml polybrene prior to spinfection (2500 rpm, 37°C, 45 min). For primary mouse T cells, plates were coated with 32 μg/ml retronectin (Takara Bio) for 24 hr before plating of $1 \times 10^6$ cells in 1 ml viral supernatant and performing a spinfection (2500 rpm, 37°C, 45 min). Viral supernatant was removed after 16 hr and replaced with RPMI supplemented with fetal bovine serum (10%; Bovogen Biologicals) and L-glutamine (2 mM; Gibco) for BW5147 cells, or mTCM for primary T cells.

## Surface IP and immunoblot analysis

$2 \times 10^7$ cells per sample were pelleted and washed twice with PBS prior to coating with 20 μg/ml polyclonal anti-mouse IgG for 45 min on ice. Cells were washed twice with PBS and lysed in 200 μl PBS/1% IGEPAL-640/P8340 protease inhibitor/10 mM iodoacetamide for 30 min on ice. Lysate was centrifuged at $20,000 \times g$ for 10 min, 10 μl of cleared lysate was taken for 5% input controls with remainder being added to 20 μl Thermo Fisher Protein G agarose beads and rotated in cold room for 2 hr. Beads were washed with lysis buffer twice then eluted with LDS and boiled. Samples were run on SDS-PAGE and transferred for blotting with 1:2000 anti-Myc primary antibody (Cell Signaling #2276) and 1:20,000 anti-mouse IgG HRP secondary (Sigma-Aldrich A0168).

## CAR T cell SKBR3 co-culture assay

$5 \times 10^4$ cells/cell line were aliquoted onto a confluent layer of SKBR3 cells in a 96-well plate at specified time points. After 8 hr, all time points were removed from plate and stained with 1:200 anti-CD69 (BioLegend #104524) on ice for 45 min. Samples were analyzed on an LSR Fortessa X20 (BD Biosciences), and data were analyzed using FlowJo v10 software.

## Flow cytometry

For CD8[+] T cell selection and transduction efficiency verification, single-cell suspensions were washed and stained with Live/Dead marker Zombie Aqua (BioLegend) for 15 min at RT in PBS, before washing and labeling for at least 30 min on ice with a panel of monoclonal antibodies (mAbs), including anti-mouse CD3ε PE (clone 145-2C11, BioLegend), anti-mouse CD8α APC-Cy7 (clone 53-6.7, BioLegend), and anti-mouse Myc-Tag Alexa Fluor 647 (clone 9B11, Cell Signaling). All samples were analyzed with an LSR II Fortessa (BD Biosciences), FACSymphony (BD Biosciences), or Aurora (Cytek), and data were analyzed using FlowJo v10 software.

## Chromium release killing assay

Standard $^{51}$Cr release assays were conducted to assess CAR T cell cytotoxicity by measuring release of radioactivity into culture supernatants as cells are lysed. Target MC57 mouse fibrosarcoma cells stably expressing human HER2 (MC57-HER2) were pre-loaded with 100 µCi $^{51}$Cr for 1 hr at 37°C, washed three times, and then $2 \times 10^4$ tumor cells were co-incubated with CAR T cells at effector-to-target (E:T) ratios ranging from 40:1 to 1.25:1. Supernatants were harvested after 4 hr of co-incubation, plated onto a 96-well scintillator-coated LumaPlate (PerkinElmer), and $^{51}$Cr release quantified using a Micro-Beta$^2$ Microplate Counter (PerkinElmer). Target tumor cells incubated in a 5% Triton X-100 solution were used as a maximum release control, while tumor cells incubated in mTCM alone were used as a spontaneous release control. Percent lysis was calculated as follows: % lysis = ((Experimental release – Spontaneous release) ÷ (Maximum release – Spontaneous release)) 100. Data in *Figures 3f and 4d* were derived from the 20:1 E:T ratio where killing was maximal for all constructs.

## IncuCyte killing assay

To measure tumor cell death over time, the live-cell imaging system IncuCyte SX3 or SX5 was used. In this assay, $8 \times 10^3$ target tumor cells per well were plated in a 96-well plate in triplicate, and the following day CD8$^+$ T cells were added at an effector to target ratio of 1:1 in mTCM media. 50 µM propidium iodide (PI; Sigma-Aldrich) was added to each well as a surrogate marker of cell death. Wells were subsequently imaged every hour for 24 hr, with phase and PI fluorescence recorded. All images were analyzed using the IncuCyte Analysis Software program, where the average PI area (µm) was calculated for each individual well from at least two images per time point. Target tumor cells incubated in a 1.2% (w/v) Saponin (Sigma-Aldrich) solution were used as a positive PI release control while tumor cells incubated in mTCM alone were used as a background PI release control for PI area calculations. Data in *Figure 6d* shows all biological replicates and time points graphed as PI area (y-axis) vs. time (x-axis) using GraphPad Prism v 9.0.0.

## Cytokine bead array

To assess cytokine secretion by CAR T cells, cytokine bead arrays on co-culture supernatants were performed. Murine CAR T cells ($1 \times 10^5$ cells) were washed once in PBS and co-incubated with either mTCM alone, a 1:1 bead-to-cell ratio of Mouse T-Activator CD3/CD28 Dynabeads (Gibco) as a positive control, nontarget MC57 parental tumor cells ($2 \times 10^4$ cells) as a negative control, or target MC57-HER2 tumor cells ($2 \times 10^4$ cells) in triplicate. After 24 hr, supernatants of co-cultures were collected and used in a LEGENDplex Mouse T Helper Cytokine Panel Version 2 Flexi Kit (BioLegend) for IFN-γ, IL-2, and TNFα, and LEGENDplex Mouse Cytokine Panel 2 Flexi Kit (BioLegend) for GM-CSF according to the manufacturer's instructions. All samples were analyzed using an LSR II Fortessa or FACSVerse (BD Biosciences) and concentration determined against a standard curve of each analyte using FlowJo v10 software.

## Confocal microscopy and cluster analysis

$8 \times 10^5$ cells were labeled with unconjugated anti-Myc primary antibody (Cell Signaling) in PBS/0.5% BSA for 30 min on ice. Cells were washed twice in PBS and further incubated with Alexa Fluor 488 anti-mouse IgG secondary antibody (Abcam) in 50 µl ice-cold RPMI for 10 min on ice. 50 µl pre-warmed RPMI was added and samples were transferred to a 37°C water bath for 10 min to induce CAR clustering. CAR clustering was halted via addition of ice-cold PBS/0.1% sodium azide. Cells were washed in PBS/0.1% sodium azide then stained on ice for 45 min with anti-CD28 APC (BioLegend) diluted in PBS/0.1% sodium azide. Cells were fixed with 3% paraformaldehyde, transferred to eight-well chamber slides (Ibidi), and stored at 4°C overnight until imaging. 3D confocal image data was collected using a Zeiss LSM 980 microscope, with 55–60 slices collected per image at a z-step size of 0.23 µm. The pinhole size used was 1 airy unit, resulting in a slice thickness of 600 nm. Image analysis was conducted using the cluster-picking function within the Imaris software package. CAR clusters (Alexa Fluor488) and CD28 clusters (APC) were counted, with the percentage of CAR clusters co-localizing with a CD28 cluster reported per cell with at least 30 cells per construct analyzed.

## In vivo tumor growth

For in vivo tumor growth analysis, $5 \times 10^5$ MC38 colon adenocarcinoma cells stably expressing human HER2 (MC38-HER2) were injected subcutaneously into the left flank of 5- to 6-week-old NSG mice and 5–6 mice were randomly assigned to each treatment group. One day later, mice were injected intravenously via the tail vein with $1 \times 10^7$ CD8$^+$ T cells transduced with the indicated CAR constructs. On days 1, 2 and 3, mice were injected intraperitoneally with $5 \times 10^4$ IU recombinant human IL-2. Mice were weighed weekly and tumors measured daily until each individual tumor reached a maximum tumor volume of 1000 mm$^3$ as per ethical guidelines, after which mice were euthanized.

## Acknowledgements

We acknowledge the use of the CSIRO Collaborative Crystallisation Centre (C3) for crystallization screening. X-ray diffraction data were collected at the MX2 beamline of the Australian Synchrotron, and we thank the beamline scientists for their technical support. Research was supported in part by the NHMRC IRIISS Infrastructure Support and Victorian State Government Operational Infrastructure Support (to WEHI).

## Additional information

### Competing interests

Assaf Elazar, Nicholas J Chandler, Jonathan Y Weinstein, Melissa J Call, Matthew E Call: Author is an inventor on a related patent WO2021229581A2. Sarel J Fleishman: Reviewing editor, eLife. The other authors declare that no competing interests exist.

### Funding

| Funder | Grant reference number | Author |
|---|---|---|
| H2020 European Research Council | Consolidator Grant 815379 | Sarel Fleishman |
| Dr Barry Sherman Institute for Medicinal Chemistry | Institutional Funding | Sarel Fleishman |
| Sam Switzer and Family | Charitable Donation | Sarel Fleishman |
| National Health and Medical Research Council | Project Grant 1158249 | Misty R Jenkins Melissa J Call Matthew E Call Sarel Fleishman |
| Harry Secomb Foundation | Charitable Donation | Matthew E Call |
| Percy Baxter Charitable Trust | Charitable Donation | Matthew E Call |
| Harold and Cora Brennen Benevolent Trust | Charitable Donation | Matthew E Call |

The funders had no role in study design, data collection and interpretation, or the decision to submit the work for publication.

### Author contributions

Assaf Elazar, Conceived and developed the proMP modelling and design methods and computed the designs, Conceived and developed the proMP modelling and design methods and computed the designs, Conceptualization, Data curation, Formal analysis, Investigation, Methodology, Software, Writing – review and editing; Nicholas J Chandler, Data curation, Formal analysis, Investigation, Methodology, Produced and purified proMP peptides and performed crystallization screening. Determined crystal structures, Produced and purified proMP peptides and performed crystallization screening. Determined crystal structures, Produced and purified proMP peptides and performed crystallization screening. Determined crystal structures, Writing – review and editing; Ashleigh S Davey, Data curation, Designed and performed in vitro primary T cell experiments and in vivo tumor control

experiments, Formal analysis, Investigation, Methodology, Writing – review and editing; Jonathan Y Weinstein, Conceived and developed the proMP modelling and design methods and computed the designs, Investigation, Methodology, Software, Writing – review and editing; Julie V Nguyen, Investigation, Produced and purified proMP peptides and performed crystallization screening. Determined crystal structures, Produced and purified proMP peptides and performed crystallization screening. Determined crystal structures; Raphael Trenker, Investigation, Methodology, Produced and purified proMP peptides and performed crystallization screening. Determined crystal structures, Produced and purified proMP peptides and performed crystallization screening. Determined crystal structures; Ryan S Cross, Methodology, Resources, Writing – review and editing; Misty R Jenkins, Funding acquisition, Methodology, Resources, Writing – review and editing; Melissa J Call, Conceived and supervised proMP peptide production and structure determination strategies. Conceived and supervised proCAR T cell design and testing strategies. Determined crystal structures, Conceptualization, Data curation, Formal analysis, Funding acquisition, Investigation, Methodology, Project administration, Supervision, Writing – original draft, Writing – review and editing; Matthew E Call, Conceived and supervised proMP peptide production and structure determination strategies. Conceived and supervised proCAR T cell design and testing strategies. Produced and purified proMP peptides and performed crystallization screening, Conceptualization, Funding acquisition, Investigation, Methodology, Project administration, Supervision, Writing – original draft, Writing – review and editing; Sarel J Fleishman, Conceived and developed the proMP modelling and design methods. Supervised the design process and bacterial expression and self-association experiments, Conceptualization, Funding acquisition, Methodology, Project administration, Software, Supervision, Writing – original draft, Writing – review and editing

### Author ORCIDs
Assaf Elazar http://orcid.org/0000-0002-5281-0908
Raphael Trenker http://orcid.org/0000-0003-1748-0517
Melissa J Call http://orcid.org/0000-0001-7684-5841
Matthew E Call http://orcid.org/0000-0001-5846-6469
Sarel J Fleishman http://orcid.org/0000-0003-3177-7560

### Ethics
All animal experiments were approved and performed in accordance with the regulatory standards of the Walter and Eliza Hall Institute Animal Ethics Committee (Approval: WEHI-2019.020).

### Decision letter and Author response
Decision letter https://doi.org/10.7554/eLife.75660.sa1
Author response https://doi.org/10.7554/eLife.75660.sa2

---

## Additional files

### Supplementary files
- Supplementary file 1. RosettaScripts and command lines.
- Transparent reporting form
- Source data 1. Uncropped gel and blot images.

### Data availability
Diffraction data have been deposited in the Protein Data Bank under the PDB accession codes 6WA0, 6W9Y and 6W9Z.

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
