## [Editor Report]

This is an interesting article that uses de novo protein design to probe the effects of oligomerization state on the activity of chimeric antigen receptors (CARS). The successful design of transmembrane domains with specific oligomeric states is an impressive result on its own. After experimentally evaluating a couple rounds of designs, the investigators settled on a design protocol that also included screening of the design candidates with docking simulations in alternative oligomerization states to check that the sequences preferred the desired oligomerization state. The designs were experimentally evaluated with gel electrophoresis and X-ray crystallography. In the end, designs that adopted well-defined dimers, trimers, or tetramers were created and carried forward in experiments as CARs.

---

## [Decision Letter]

**Decision letter after peer review:**

Thank you for submitting your article "de novo designed transmembrane domains tune engineered receptor functions" for consideration by *eLife*. Your article has been reviewed by 3 peer reviewers, and the evaluation has been overseen by Nir Ben-Tal as Reviewing Editor and José Faraldo-Gómez as the Senior Editor. The following individuals involved in review of your submission have agreed to reveal their identity: Brian Kuhlman (Reviewer #1); Dieter Langosch (Reviewer #2); James J. Chou (Reviewer #3).

Essential revisions:

1. in vitro experiments showed a clear correlation between oligomerization state of the CAR and cytokine secretion when CAR T cells were exposed to HER2+ cancer cells. The higher order oligomers also were more effective at slowing tumor growth in mice injected with HER2 tumor cells. These results confirm previous observations that dimeric CARs (via disulfide formation) are more effective than monomeric CARs. One exciting finding was that the designed CAR tetramer was as effective at suppressing tumor growth in mice as a standard CAR construct used in the field (transmembrane domain derived from CD28), but the tetramer CAR stimulated less cytokine release than the CD28TM CAR in vitro. The CAR T therapies currently used in the clinic frequently stimulate dangerous levels of cytokines, if a CAR T cell can be created that is as effective as current treatments but overall cytokine release is lowered, this could be an improvement over current treatment options. One caveat about the efficacy data presented in this paper is that CAR T cells were administered to the mice only a single day after injection of the cancer cells. More rigorous tests of efficacy will be needed to determine if the tetrameric CAR is on par with standard constructs used in therapy.

2. The protein design process contained no consideration for how activation signals are transmitted from the extracellular domain (ECD) of the CAR to the intracellular activation domains. Does this suggest that specific conformational changes are not a part of the activation process? It would be great if the authors could comment on this. Do the results say anything about how binding to the ECD does or does not activate signaling?

3. TMH oligomer design. It appears that the crystal structures and the designed structures agree with rather high success rate. Wouldn't you be able to predict the oligomeric structure of a given TMH sequence? Can your design protocol be revised to predict the assumed dimeric structure of CD28 TMH? We are asking because there is no direct biochemical evidence that the CD28 TM mutations have abolished TMH dimerization.

4. A potential concern of analyzing TMD oligomeric state using SDS-PAGE is that the strong assembly interactions can survive the harsh condition of SDS-PAGE but the weak ones don't. You should comment on the possible issue, if any, of TMD forming higher-order clusters in the membrane in the discussion. Has this issue been eliminated in the design phase?

5. An observation of profound impact is the beautiful linear correlation between the proCAR TMD oligomeric state and tumor killing in vivo. However, the increase is not obvious for in vitro cell cytotoxicity assay. Did the two studies use two different cancel cell lines? Please explain.

6. For readers not working on CAR-T, can you discuss why heterotypic mixing of CAR-CD28TM and endogenous CD28 via TM-TM dimerization can cause much greater cytokine release given both contain the same CD28 signaling tail? Is it because CAR-CD28TM mixing with CD28 would increase the total amount of CD28 that can engage HER2 and activate? If so, does the 2-6 fold increase in cytokine release make qualitative sense? Do CAR and endogenous CD28 induce different cytokine release?*Reviewer #1:*

This is an interesting paper that uses de novo protein design to probe the effects of oligomerization state on the activity of chimeric antigen receptors (CARS). The successful design of transmembrane domains with specific oligomeric states is an impressive result on its own. The proteins were designed using rotamer-based sequence optimization in Rosetta with an energy function specific for the membrane environment. During the design process it was important to explicitly reward sequence diversity as low diversity sequences (i.e. many leucines) produced the lowest energies when evaluated on the target backbone, but showed little specificity for a single conformation when docking simulations were performed with the designs. After experimentally evaluating a couple rounds of designs, the investigators settled on a design protocol that also included screening of the design candidates with docking simulations in alternative oligomerization states to check that the sequences preferred the desired oligomerization state. The designs were experimentally evaluated with gel electrophoresis and X-ray crystallography. In the end, designs that adopted well-defined dimers, trimers, or tetramers were created and carried forward in experiments as CARs.

In vitro experiments showed a clear correlation between oligomerization state of the CAR and cytokine secretion when CAR T cells were exposed to HER2+ cancer cells. The higher order oligomers also were more effective at slowing tumor growth in mice injected with HER2 tumor cells. These results confirm previous observations that dimeric CARs (via disulfide formation) are more effective than monomeric CARs. One exciting finding was that the designed CAR tetramer was as effective at suppressing tumor growth in mice as a standard CAR construct used in the field (transmembrane domain derived from CD28), but the tetramer CAR stimulated less cytokine release than the CD28TM CAR in vitro. The CAR T therapies currently used in the clinic frequently stimulate dangerous levels of cytokines, if a CAR T cell can be created that is as effective as current treatments but overall cytokine release is lowered, this could be an improvement over current treatment options. One caveat about the efficacy data presented in this paper is that CAR T cells were administered to the mice only a single day after injection of the cancer cells. More rigorous tests of efficacy will be needed to determine if the tetrameric CAR is on par with standard constructs used in therapy.

One thing that struck me is that the protein design process contained no consideration for how activation signals are transmitted from the extracellular domain (ECD) of the CAR to the intracellular activation domains. Does this suggest that specific conformational changes are not a part of the activation process? It would be great if the authors could comment on this. Do the results say anything about how binding to the ECD does or does not activate signaling?*Reviewer #2:*

The authors designed and computationally refined a set of novel self-associating TM helices which were shown to form dimers and trimers by X-ray crystallography. The dimer (Car2) and trimer (Car3) as well as a modelled tetramer (Car4) were then fused to extracellular antigen-recognition and intracellular signalling domains. They were then compared to a monomer (Car1) and the original CD28 TMD in their ability to kill cancer cells in vitro and in vivo and to release various cytokines. Also, the authors show that a specific amino acid motif within the wild-type CD28 TMD of the CD28 CAR mediates interaction with the endogenous CD28 and that this is responsible for cytokine release, an undesired side reaction associated with cancer cell killing. Since their CARs with de novo designed TMDs show much reduced cytokine release, this confirms their notion that non-natural TMDs would isolate CARs from endogeneous CD28. The novel functional aspect about the designer CARs presented here is thus that cytokine secretion is less troubling when compared with the original CD28 CAR28. Most interestingly, they find that the antitumor activity of the constructs tested in an engineered mouse tumour model as well as induced cytokine release scales with the oligomeric state.

In sum, the work is not only very elegant from a membrane protein engineering point of view. Rather, it represents a fine example of how protein design can be translated into medical applications.

*Reviewer #3:*

The paper by Elazar et al., describes a highly significant and rigorously performed study that addresses the influence of TMD-mediated chimeric antigen receptor (CAR) oligomerization on the antitumor activity and cytokine release of CAR-T cells.

The transmembrane helices (TMHs) of most single-pass transmembrane receptors have long been considered as mere membrane anchors, and hence their function in receptor signaling has often been neglected. In recent years, however, several studies have found that TMH oligomerization can play an active and often essential role in the signaling mechanism of receptors including growth factor receptors, death receptors, and immune receptors. In this study, the authors have demonstrated that designed TMH oligomerization, which mediates CAR oligomerization, can have profound impact on the activity and cytokine release of CAR-T cells. Their findings have illuminated another dimension for CAR-T/NK engineering and optimization.

Specifically, the authors first performed de novo design of completely new TMHs that can form stable parallel dimer, trimer, or tetramers, as validated by SDS-PAGE and crystallography. Then, by replacing the TMH of the most used CARs, which is the TMH of CD28, with the designed TMHs, they found that the antitumor activity of the CAR-T cells in mice scaled linearly with CAR oligomeric state encoded by the designed TMH. More strikingly, their in vitro assays showed that the CARs with the designed TMHs all induced 2-10-fold less cytokine release than the CAR with CD28 TMH, raising the suspicion that the CD28 TMH may pair CAR with endogenous T cell CD28 leading to higher level of CD28 signaling independent of the CARs. Indeed, they have shown that a set of mutations in the CD28 TM of the regular CAR that is likely to disrupt TMH homodimerization led to 2-6-fold reduction in CAR-T cell cytokine release. Collectively, their results suggest that there is ample room for improving the CAR-T antitumor activity / cytokine release ratio by optimizing the CAR TM sequence, and it remains to be seen if this approach can be used to achieve better outcomes in higher animals.

---

## [Author Response]

Essential revisions:1. In vitro experiments showed a clear correlation between oligomerization state of the CAR and cytokine secretion when CAR T cells were exposed to HER2+ cancer cells. The higher order oligomers also were more effective at slowing tumor growth in mice injected with HER2 tumor cells. These results confirm previous observations that dimeric CARs (via disulfide formation) are more effective than monomeric CARs.

This is an important point and we had not explicitly acknowledged that dimer formation was already thought to be important for optimal CAR function in the original manuscript. We have clarified this in the Introduction (lines 74-75) with relevant references.

One exciting finding was that the designed CAR tetramer was as effective at suppressing tumor growth in mice as a standard CAR construct used in the field (transmembrane domain derived from CD28), but the tetramer CAR stimulated less cytokine release than the CD28TM CAR in vitro. The CAR T therapies currently used in the clinic frequently stimulate dangerous levels of cytokines, if a CAR T cell can be created that is as effective as current treatments but overall cytokine release is lowered, this could be an improvement over current treatment options. One caveat about the efficacy data presented in this paper is that CAR T cells were administered to the mice only a single day after injection of the cancer cells. More rigorous tests of efficacy will be needed to determine if the tetrameric CAR is on par with standard constructs used in therapy.

We agree with the Reviewer that this is an important caveat. The MC38-HER2 tumor model was used here because it is relatively rapid and it is well suited to quantitative comparisons of relative functional potency, but the extremely aggressive growth of MC38 necessitates very early CAR T cell administration in order to observe tumor control. The revised Discussion (lines 475-478) now makes it clear that testing the clinical potential of proCARs will require additional experiments in models that more closely approximate treatment of cancer patients.

2. The protein design process contained no consideration for how activation signals are transmitted from the extracellular domain (ECD) of the CAR to the intracellular activation domains. Does this suggest that specific conformational changes are not a part of the activation process? It would be great if the authors could comment on this. Do the results say anything about how binding to the ECD does or does not activate signaling?

This is a very good point. We can’t rule out dynamics or conformational changes in response to ligand binding to the ECD driving the CAR-T signaling, but as the Reviewer notes, our designs don’t consider such possibilities yet increase activation. The revised Discussion (lines 452-458) explains that our results, particularly the linear correlation between oligomeric state and activation, favor the view that the specific geometry of interaction and conformational changes or dynamics play a minor role if any in receptor activation.

3. TMH oligomer design. It appears that the crystal structures and the designed structures agree with rather high success rate. Wouldn't you be able to predict the oligomeric structure of a given TMH sequence? Can your design protocol be revised to predict the assumed dimeric structure of CD28 TMH? We are asking because there is no direct biochemical evidence that the CD28 TM mutations have abolished TMH dimerization.

Rather than rely on an independent model of this interface, we based our approach on Tox-Luc data from Leddon et al., (Front Immunol 2020, cited in the manuscript) showing that the Y-T interactions drive TMD dimerisation in a manner very similar to CD3 zeta. We, therefore, used the latter structure as a template for introducing the mutations to the CD28 TMD. We agree that the results do not provide “proof” that CD28 associates through the interface observed in the CD3 zeta structure. However, the fact that the mutations were isosteric, yet abolished association and reversed the amplification of T cell functions, is evidence in favour of this view.

4. A potential concern of analyzing TMD oligomeric state using SDS-PAGE is that the strong assembly interactions can survive the harsh condition of SDS-PAGE but the weak ones don't. You should comment on the possible issue, if any, of TMD forming higher-order clusters in the membrane in the discussion. Has this issue been eliminated in the design phase?

The reviewer is correct to point out that SDS-PAGE migration does not report on weak self-association and therefore does not rule out unintended secondary interfaces that could drive higher-order interactions in the membrane. We now clearly state this is the revised Discussion (lines 420-423). We do note, however, that the striking linearity in the correlations between the designed proMP oligomeric state and the measured proCAR functional output suggests that the designed interfaces drive the dominant interactions, and we see no evidence of influence by unintended higher-order assemblies. Indeed, the design process favors the desired oligomeric state (up to tetramers) but we would not think that computational design could “guarantee” that higher-order clusters were eliminated.

5. An observation of profound impact is the beautiful linear correlation between the proCAR TMD oligomeric state and tumor killing in vivo. However, the increase is not obvious for in vitro cell cytotoxicity assay. Did the two studies use two different cancel cell lines? Please explain.

We agree that the lack of scaled activity when in vitro cytotoxicity is used as a measure is striking. The revised manuscript addresses this directly in the Results section (lines 369-371) and offers a possible explanation. Others (see Majzner et al., 2020, cited in the manuscript) have observed that CARs with apparently equivalent killing activity against high-antigen targets can display very different activities against low-antigen targets in the same assay, suggesting that in vitro killing is easily saturated at high antigen densities. The cell lines used for the in vitro (MC57 mouse fibrosarcoma cells) and in vivo (MC38 mouse colon carcinoma cells) experiments are both engineered to express high levels of HER2 and we have no evidence that the difference cell lines is the source of this discrepancy. We believe that the oligomeric proCARs segregate in vivo but not in vitro because in vivo tumor control over days to weeks is a function of multiple parameters that are all affected by CAR signal strength (T cell proliferation, survival, cytokine production and cytotoxic potential) whereas short-term in vitro killing primarily measures release of cytotoxic effector molecules.

6. For readers not working on CAR-T, can you discuss why heterotypic mixing of CAR-CD28TM and endogenous CD28 via TM-TM dimerization can cause much greater cytokine release given both contain the same CD28 signaling tail? Is it because CAR-CD28TM mixing with CD28 would increase the total amount of CD28 that can engage HER2 and activate? If so, does the 2-6 fold increase in cytokine release make qualitative sense? Do CAR and endogenous CD28 induce different cytokine release?

We agree that this is especially complicated for the non-specialist and have clarified in the revised Discussion (lines 433-437) that we believe the effect of the CD28 TMD interaction on CAR signaling to be a direct consequence of recruitment of additional costimulatory signaling tails into activated receptor complexes: each CD28 TMD in a CAR dimer could theoretically recruit an endogenous CD28 dimer via domain-swapped TMD interactions. While this is entirely consistent with several-fold amplification of cytokine output by the CD28 TMD, it is not known whether signaling from a CD28 tail in the context of a CAR is qualitatively or quantitatively equivalent to that in the context of the native protein. Thus we hesitate to draw direct connections between the stoichiometry of the interaction and the observed fold-differences in cytokine release.